# GriNNder: Large-Scale Full-Graph Training of Graph Neural Networks on a Single GPU with Storage

## Abstract

Full-graph training of graph neural networks (GNNs) processes the entire graph at once, preserving all input information and enabling straightforward validation of algorithmic gains. However, it typically needs multiple GPUs/servers, increasing costs and inter-server communication. Although single-server methods reduce expenses, they remain constrained by limited GPU/host memory as graph sizes grow. Furthermore, naïvely applying storage-based methods from other domains to mitigate such a limit is infeasible for handling large-scale graphs. Here, we introduce *GriNNder*, the first storage-based framework (e.g., using NVMe SSDs) for scalable and efficient full-graph GNN training. GriNNder alleviates GPU memory bottlenecks by offloading data to storage, while keeping read/write traffic to and from the storage device minimal. To achieve this, from the observation that cross-partition dependencies follow a power-law distribution, we introduce an efficient partition-wise caching strategy for handling intermediate activations/gradients of full-graph dependencies with host memory. Also, we design a regathering mechanism for the gradient engine that minimizes storage traffic and propose a lightweight partitioning scheme that overcomes the memory limitations of existing methods. GriNNder achieves up to $9.78\times$ speedup over the state-of-the-art baseline and comparable throughput to distributed baselines while enabling previously infeasible large-scale full-graph training with a single GPU.

## 1 Introduction

Graph neural networks (GNNs) are powerful tools for learning from graph-structured data, applicable to social networks [23], protein analysis [27], and even classic vision tasks [13]. Since graphs can represent almost any unstructured relationship, GNNs hold broad potential across diverse domains.

Most GNNs are trained with full-graph or mini-batch training. Full-graph training [90, 89, 42, 25, 79, 72] iteratively processes the entire graph information, which simplifies identifying algorithmic gains. However, this requires storing all intermediate activations/gradients, which can easily overflow GPU memory. While scaling GPUs is an option, it incurs significant hardware costs/communication overhead, often leading to poor efficiency. Mini-batch training [11, 14, 33, 99] utilizes graph sampling to resize the input to fit GPU memory capacity. However, it often results in information loss (e.g., neighbors' features). Moreover, it also requires extensive tuning of sampling strategies and hyperparameters, which complicates finding the optimal performance of the developers' algorithms [6].

Aforementioned limitations, which come from the hardware constraints, hinder researchers from developing their algorithms flexibly. Our survey on recent conference GNN papers (Appendix A) confirms the appeal of full-graph training for its simplicity and fidelity. Around half of them opted for full-graph training, but many of them reported out-of-memory with large graphs. To address this, some single-server full-graph training methods [97, 92] have been proposed, but suffer from

GPU/host memory limit as graph sizes grow (Appendix B). Thus, we devise a novel approach that enables full-graph training of large graphs under limited resources (i.e., a single GPU) with storage (e.g., NVMe SSD).

One might think that existing storage-based solutions can compensate for limited GPU and host memory. However, such solutions have fundamental limitations and cannot be directly applied to full-graph GNN training. For instance, in the context of large language models (LLMs), several solutions utilize storage [74, 78] by offloading weight parameters/optimizer states to NVMe devices. Unlike LLMs, which typically have large weights and hence large optimizer states, GNN weights are shared among all vertices, with only a few (e.g., 2-5) layers. This indicates a need for offloading vertex activations/gradients instead, but this brings a non-trivial challenge of addressing the complicated dependency (i.e., edges) between layers. These dependencies cause frequent random accesses, which put a significant I/O burden on the channel between the GPU and storage.

In the case of mini-batch GNN training, storage-based methods [70, 88, 59, 44] primarily focus on efficiently constructing mini-batches while leveraging storage to hold initial graph-related features. However, extending storage-based mini-batch training [70, 88, 59, 44] to full-graph training (called micro-batch training [97]) also faces the limitations because it only focuses on handling initial features (not intermediate activations/gradients), and further suffers from the GPU out-of-memory due to neighbor explosion (Appendix C).

Specifically, following are three key challenges when employing storage for full-graph GNN training:

1. *Storage I/O Bottlenecks*: Despite the improved bandwidth of NVMe SSDs, they are far slower than host memory and suffer from inefficient I/O due to the storage page granularity.
2. *Data Amplification*: Existing methods [71, 26, 92] utilize activation snapshots to enable sequential storage access to activations. However, this approach becomes impractical when employing storage, since it inflates memory usage and I/O traffic.
3. *Impractical Partitioning*: We need to iteratively conduct graph partitioning until the required memory size is met to fit the GPU memory size. However, since existing approaches [97, 92] rely on standard partitioning algorithms [47, 49, 53], they often exceed host memory limits with large graphs, requiring a separate large-memory cluster/server.

Here, we introduce *GriNNder*, the first framework enabling fast full-graph GNN training under tight resources, using an NVMe SSD and a single GPU. It tackles the above challenges with the following:

- Partition-aware graph caching: From the observation that the cross-partition dependencies also follow a power-law distribution, we exploit this characteristic and utilize host memory as an efficient partition-wise cache with optimized I/O policies, minimizing inefficient storage I/O.
- Grad-engine activation regathering: A method to minimize redundant data storage in the automatic gradient computation engine, mitigating the data amplification in the existing offloading solutions.
- Switching-aware partitioning: A fast, memory-efficient partitioning algorithm for limited-resource settings, avoiding the high memory footprint of standard partitioners.

We implemented GriNNder as `PyGriNNder`, allowing users to easily utilize PyTorch Geometric [26] code by inheriting the model class. Notably, GriNNder does not alter any of the model/training algorithm, ensuring seamless migration without the risk of accuracy drop. Experiments show that GriNNder achieves throughput comparable to distributed baselines and up to 9.78× speedup over the state-of-the-art, enabling previously infeasible large-scale graph training only with a single GPU.

## 2 Background: full-graph GNN training

Figure 1 shows full-graph training of a two-layer GNN on a toy graph depicted in Figure 1a. From the topology, the two-layer dependency can be drawn as in Figure 1b. Starting from the input features denoted with circled vertex ids, the features are passed by *message passing* to the features of destination vertices in the intermediate layer. The message passing of the second layer proceeds with the same dependency, which creates the output embeddings for the vertices.

Figure 1c illustrates the typical layer-by-layer procedure of conducting full-graph training on the GNN. To compute an output feature vector of a vertex, the features of source vertices from the previous layers need to be *aggregated* (e.g., average). For example, vertex feature ⓐ has dependencies from vertex features ⓐ, ⓑ, and ⓖ, including an implicit self-directed edge. Similarly, vertex feature ⓘ

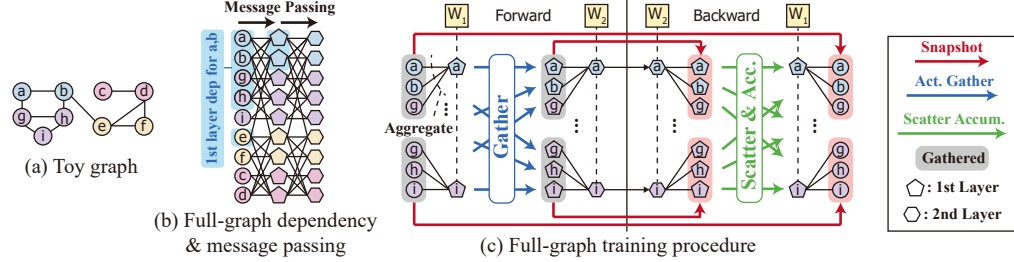

Figure 1: Example full-graph training procedure with a two-layer GNN.

has dependencies from vertex features ⓖ, ⓗ, and ⓘ. After the aggregation, multiplying them with the shared weight matrix (i.e., $W_1$) followed by misc operations such as normalization and activation produces the final output features for the layer (denoted by ⬠). For the next layer, the output features are *gather*ed to make inputs for aggregation under the same dependency (blue arrows). The gathered activations are saved (i.e., snapshots) in the GPU/host memory for later use in the backward pass.

In the backward, the dependency is inverted, where the output of vertex feature/gradient ⟨a⟩ is delivered to ⟨a⟩, ⟨b⟩, and ⟨g⟩ to compute their gradients. For this, the previously memory-stored snapshots are loaded (red arrows), and the computed gradients of the corresponding source vertex features are scatter-accumulated to the vertices of the previous layer (green arrows).

For workloads that fit on a GPU memory, this procedure ensures fast training by utilizing massive parallelism and high memory bandwidth. However, this comes at the cost of capacity pressure, because the entire GNN with all its intermediate data has to fit within GPU memory. A straightforward solution is to scale out [85, 72], but it often suffers from high system cost and slow network throughput.

To address such issues, several methods targeting tight resource constraints (i.e., limited GPU memory capacity) have been proposed [97, 92]. However, they still suffer from GPU/host memory limit and impractical partitioning, which are discussed in Appendix B. Also, extending storage-based mini-batch training to full-graph training faces GPU memory limit due to neighbor explosion (Appendix C). On the other hand, GriNNder addresses such issues by employing storage with efficient strategies.

## 3 Full-graph GNN training workflow with storage employment

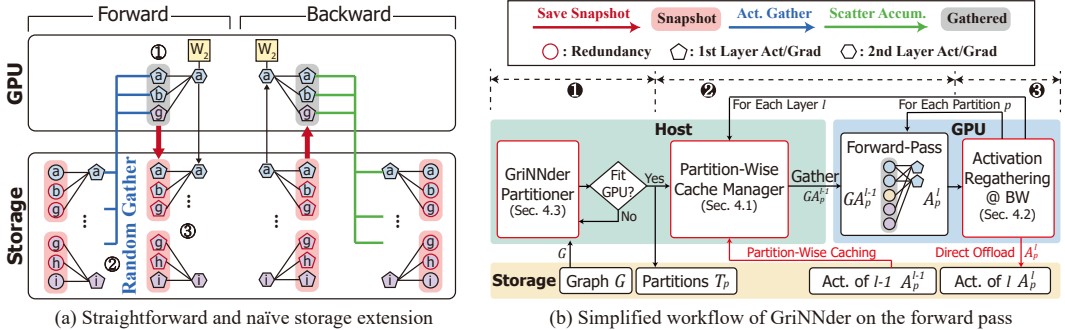

Figure 2: Workflow of GriNNder compared to the naïve storage extension of full-graph training.

Given the full-graph training from Figure 1, a straightforward method would be to place the small weights (and their gradients) on the GPU and the large activations (and their gradients) on the storage. Figure 2a illustrates an example procedure for processing a single vertex, ⟨a⟩. Since the neighbors of ⟨a⟩ (⟨a⟩, ⟨b⟩, and ⟨g⟩) are small enough to fit within the GPU memory, the training can be conducted.

However, this approach yields sub-optimal performance for three main reasons: ① It is non-trivial to ensure that the neighbors of a target vertex are small enough to fit within GPU memory, which is necessary for enabling full-graph GNN training on a single GPU. ② Gathering the feature vectors of ⟨a⟩, ⟨b⟩, and ⟨g⟩ requires random reads from storage. Since storage devices typically operate at page granularity (e.g., 16 KiB), such random access leads to substantial inefficiencies. ③ While the existing snapshot feature in PyTorch [71] and an existing method [92] enables sequential access patterns, it introduces significant *redundancy*, resulting in inflated write traffic. For instance, ⟨g⟩ appears redundantly in the snapshots of all its neighboring vertices—⟨a⟩, ⟨h⟩, and ⟨i⟩.

To address the above limitations, we propose GriNNder, the first framework that enables storage-offloaded full-graph GNN training in environments with limited GPU and host memory. The overall workflow of GriNNder is illustrated in Figure 2b (see Appendix D for the full detailed algorithm).

To mitigate the above three challenges, we introduce solutions for each. ❶ Before training begins, the entire graph $G$ is partitioned into small graphs denoted $T_p$ such that its activation $A_p^l$ and its dependency activations $GA_p^{l-1}$ for layer $l$ fit in the GPU memory. This partitioning procedure needs to be iteratively conducted until an adequate number of partitions is found to fit such memory usage to the GPU. Thus, we propose a lightweight partitioning method, which is suitable in limited environments (Section 4.3). ❷ GriNNder iterate over each $T_p$ for every layer, loading the corresponding input activation $GA_p^{l-1}$, computing $A_p^l$, and writing the result to storage. To avoid fine-grained access patterns, the host-memory cache is managed at the *partition granularity*. Furthermore, to minimize storage I/O traffic, the host memory *caches* the input activations required for the current layer's computation (Section 4.1). ❸ GriNNder redesigns the gradient engine to regather input activations ($GA_p^{l-1}$) on demand, rather than redundantly snapshotting them (Section 4.2).

# 4 GriNNder design

## 4.1 Partition-aware graph caching

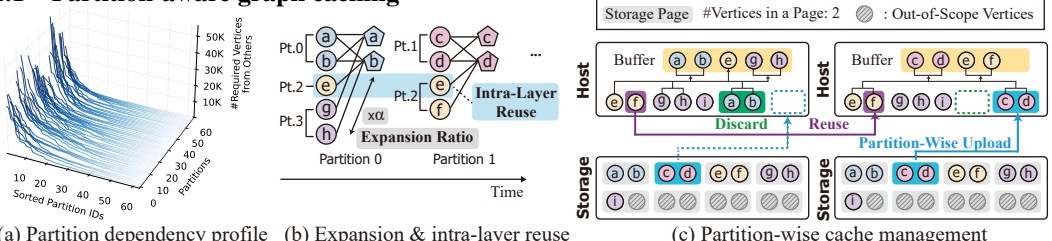

(a) Partition dependency profile    (b) Expansion & intra-layer reuse    (c) Partition-wise cache management

Figure 3: Details and rationales of partition-aware graph caching. `Pt.1` denotes the `Partition 1`.

*Key takeaway: Similar to the behavior of vertices in a graph, cross-partition dependencies also follow a power-law distribution. We can exploit this characteristic to address inefficient storage I/O.*

We observe that, just as vertex degrees in real-world graphs typically follow a power-law distribution, cross-partition access patterns exhibit similar behavior. This arises from the well-known tendency of real-world graphs to form clusters [55]. Figure 3a shows that such characteristics exist in practice. It shows statistics from each partition of the IGBM dataset (represented on the $y$-axis as `Partitions`). For each partition, we count how many vertices are required from other partitions. These counts are sorted along the $x$-axis (`Sorted Partition IDs`). The plot shows that out of 64 partitions, most of the dependencies are confined to ~10 partitions (see Appendix E for more datasets). From this observation, we design the following two key mechanisms.

**Layer-wise partition caching**: Within a layer, many partitions share activations/gradients. For instance, in Figure 3b, vertex $e$'s activation is used in both partitions 0 and 1. When the average expansion ratio (#required/#target) is $\alpha$, the activations are reused $\alpha - 1$ times on average within that layer. This leads to redundant data accesses to the storage device. To mitigate this inefficiency, we introduce a strategy called *intra-layer reuse*, where frequently reused partitions within a GNN layer are cached in host memory. For the other data that have less or no intra-layer reuse (e.g., graph topology/output activations), we choose to bypass the host memory with GPUDirect Storage [67] (GDS). This has the effect of reducing the I/O traffic and avoiding cache conflict at the host side. Please note that GriNNder can be generally used even when GDS is unavailable (see Appendix T).

**Partition-wise cache management**: To support the aforementioned feature, GriNNder uses a partition as the load/evict granularity for the host-memory caches. One naive alternative would be to load/evict at a vertex granularity. However, in this way, every time a cache miss occurs, reading a single vertex feature (64~1,024B) from the storage device is needed. Since storage devices access data at a page granularity (e.g., 16KiB), this would incur a substantial amount of unnecessary I/O. In contrast, loading and evicting at a partition granularity alleviates such overhead, because the size of a partition is typically a few GBs. For example, processing partition 0 (vertices $a$ and $b$) has dependencies to vertices $a$, $b$, $e$, $g$, and $h$, which loads three partitions to the host memory: 0, 2, and 3 as illustrated in Figure 3c. Then, when it proceeds to partition 1, it has dependencies to $c$, $d$, $e$,

and $f$ that span over partitions 1 and 2. For this, we reuse partition 2, which is already cached in memory. For loading partition 1, we evict an unused partition (partition 0 in this example, assuming there is not enough host memory space). This way, the vertex features are reused without causing fine-grained random accesses to the storage. In the worst case, partition-wise management could only cause overhead if the dependencies are uniformly spread over many partitions. However, in the above observation, the dependencies of a partition are confined to only a few other partitions. Therefore, it can show stable caching performance. For the detailed comparison between the partition-wise and vertex-wise management, see Appendix F. We also minimize the latency by overlapping processing and cache management, and maximizing the sequential access in the GPU (Appendix G).

**Detailed procedure.** Figure 4 illustrates the brief forward/backward procedure with more details.

Figure 4a depicts the forward pass for partition 0 of layer 1. ① Layer 0's activations ($A0$) are loaded into the host-memory cache at the partition granularity. ② The partitioned graph structure $T_0$ (topology 0) is uploaded directly to the GPU. ②' The required vertex activations $GA0$ is sent from the cache to the GPU. ③ The GPU executes the forward pass to output the next activation $A1$. ④ Computed activations $A1$ are offloaded to storage via GDS as they are not needed again for the current layer. We skip the snapshot to reduce redundancy (Section 4.2).

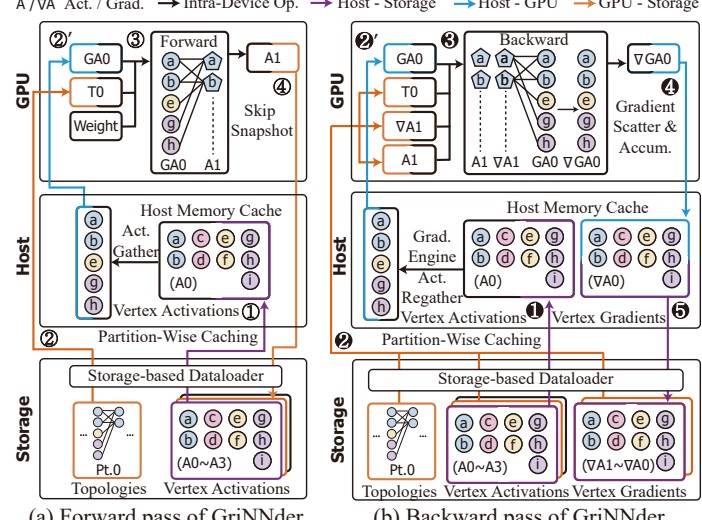

Figure 4b further illustrates the backward for the same partition of layer 1. The procedure mirrors the forward, but in reverse order, with slightly added complexity from activation gradients ($\nabla A1$ and $\nabla GA0$). ❶ Similar to the forward, the activations ($A0$) are cached in host memory partition-wise for frequent reuse. ❷ In the backward pass, the activations $A1$ and activation gradients $\nabla A1$ have to be directly loaded from the storage, in addition to the partitions. Unlike the forward, whose objective is to compute $A1$, the backward takes $A1$, $\nabla A1$, and $GA0$ as inputs and produces $\nabla GA0$. ❷' Again, similar to the forward, $GA0$ is fetched from the host memory cache through regathering, not from snapshots (Section 4.2). ❸ Using the loaded activations/gradients, the GPU computes the activation gradients ($\nabla GA0$). ❹ The source vertices' gradients ($\nabla GA0$) are updated in host memory with a scattered accumulation, ensuring correctness for vertices shared across partitions. During this, host memory works as a write-back buffer for vertex activation gradients. ❺ Once the entire layer is processed, gradients are offloaded to the storage.

## 4.2 Grad-engine activation regathering

*Key takeaway: PyTorch has limitations in supporting the aforementioned partition-wise cache management during training, particularly due to its requirement to store redundant snapshots.*

One of the key challenges for employing storage in full-graph GNN training is *data amplification*, where repeated snapshots of input activations inflate both memory capacity and storage I/O demands. As described in the previous subsection, the strategy of GriNNder is to partition the graph and cache graph features/gradients in the host memory at the partition granularity.

Unfortunately, the autograd engine of existing frameworks such as PyTorch's `torch.autograd` [71, 26] is not designed with such optimizations, and requires a significant amount of host memory when employing offloading, as drawn in (Figure 5a). The vanilla autograd engine stores activation snapshots ('Snap.') and intermediate snapshots of all operations ('Intermed.'), such as normalization ($I0$), and activation function ($I0'$). While this is a reasonable design for vision or language models, it struggles on GNNs with limited memory capacity and huge activation sizes.

Figure 4: GriNNder forward/backward for layer 1.

(a) Forward pass of GriNNder.  (b) Backward pass of GriNNder.

To mitigate this limitation, we introduce *grad-engine activation regathering*, illustrated in Figure 5, which eliminates these inefficiencies. First, the activation snapshot $GA0$ is essentially a reorganization of the activations $A0$. Based on this observation, we *regather* the activation just in time to build $GA0$ each time they are needed. This removes the unnecessary time and memory space for activation snapshots. While this adds some extra regathering overheads at the host, storing all the snapshots would easily overflow the memory, increasing the storage bandwidth demand. Second, the intermediate values are also

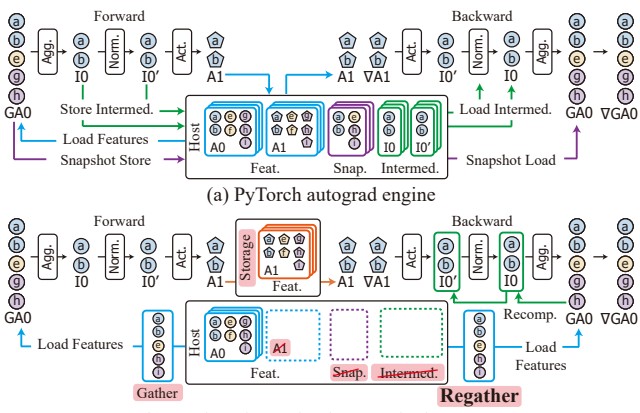

(a) PyTorch autograd engine

(b) Grad-engine activation regathering (Ours)

Figure 5: Advantages of grad-engine activation regathering

removed from the host memory and recomputed just in time from the regathered $GA0$. In the figure, $I0$ is recomputed by aggregation using the topology, and $I0'$ is obtained by further applying normalization. This is analogous to checkpointing techniques [92, 12]. Finally, the output feature $A1$ is removed from the memory by bypassing the host memory and is directly written to the storage. We further compare it with another method [92] in Appendix H.

**I/O volume and memory footprint.** Let $D = |V||H|$. During the forward on a layer, the baseline autograd engine consumes $(2\alpha + 3)D$ traffic between the GPU and the host, for the snapshots $(2\alpha D)$, intermediate values $(2D)$, and outputs $(D)$. Since the baseline easily exceeds the host memory limit, it mandates the employment of OS swap memory with storage, and most of that traffic becomes the traffic between the GPU/host and the storage. GriNNder only consumes $\alpha D$ between the GPU and the host, $D$ between the GPU and the storage, and $D$ between the host and the storage while caching (when only cold misses exist). In other words, while the baseline suffers from huge and slow storage traffic proportional to $\alpha$, grad-engine activation regathering only requires a $2D$ amount of storage traffic. In terms of the memory footprint during the forward, the baseline stores snapshots $(\alpha D)$, activations $(D)$, and intermediate values $(2D)$ per layer. On the other hand, grad-engine activation regathering only occupies $D$ space on the host memory, and $D$ on the storage for the outputs without redundancy. For more in-depth analyses with another baseline [92], please refer to Appendix I.

### 4.3 Switching-aware partitioning

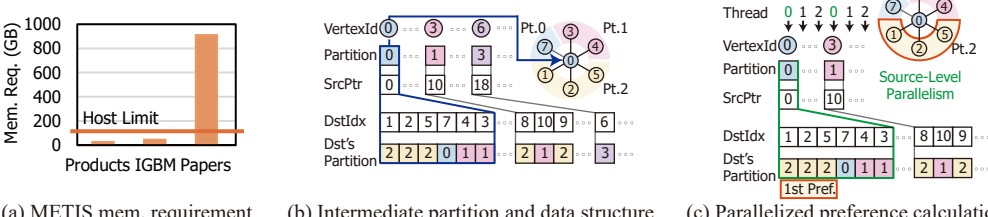

(a) METIS mem. requirement    (b) Intermediate partition and data structure    (c) Parallelized preference calculation

Figure 6: Motivation and a high-level overview of switching-aware partitioning.

*Key takeaway: Existing partitioning algorithms (e.g., METIS-based) often incur significant hardware costs, making them impractical for GNN training in resource-constrained environments.*

Graph partitioning is an important enabler that allows GriNNder to efficiently utilize GPU memory and manage caches with minimal storage bandwidth demand. Although existing partitioners used in GNN domains (e.g., METIS-based [47, 49, 53, 91, 60]) output near-optimal partitions, they often exceed single-server memory limits (Figure 6a) for large datasets such as Papers [38] (Appendix J). If partitioning has to be performed on external servers for this, it breaks the purpose of training GNNs on a single machine/GPU. Since the partitioning needs to be repeatedly iterated to find the adequate number of partitions to fit in the GPU, this issue is critical. This clearly shows the need for a lightweight partitioning method.

Inspired by streaming partitioning (Spinner [63], opted for distributed cloud systems), we devise a lightweight switching-aware partitioning, which has low memory consumption and is suited for

GriNNder. The key is to minimize the use of auxiliary data structure, whose size often largely surpasses that of the graph itself. From an arbitrary partition, we iteratively refine them to reduce the number of dependent partitions until convergence. Detailed procedures and design insights are provided in Appendix K, and we illustrate the brief overview in the following paragraphs.

Figure 6b outlines an example intermediate state along with the data structure. The data structure follows the compressed sparse row (CSR), which comprises source pointers (`SrcPtr`) and destination indices (`DstIdx`). On top of this, we manage another array (`Dst's Partition`) and fill this with the partition IDs of each destination index in `DstIdx`. In Figure 6b, the vertex 0 has neighbors of vertex $\{1, 2, 5, 7, 4, 3\}$. For each neighbor, we fill the `Dst's Partition` with its partition $\{2, 2, 2, 0, 1, 1\}$.

In a high-level view, the algorithm attempts to move to the partition with the most neighbors to reduce the number of dependent partitions, while keeping the sizes of partitions similar. In Figure 6b, the vertex 0 prefers the partition 2 (denoted by '1st Pref.' in Figure 6c) because its neighbors are mainly placed in partition 2. We search for such preferences on each source vertex (called source) in a multi-threaded manner and move each source vertex to the preferred partition as illustrated in Figure 6c. By updating the preferred partitions iteratively following the above procedure until convergence, we can minimize the average expansion ratio ($\alpha$) of the partitions. In addition, we also consider the balance between the partitions along with second-order preferences. For detailed decision-making and parallelized preference calculation and partition update, please refer to Appendix K.

**Memory Efficiency**: Switching-aware partitioning uses only a standard CSR representation— `SrcPtr`, `DstIdx`—plus a `Dst's Partition` array to record each neighbor's current partition. This totals $\mathcal{O}(|V| + |E|)$ space, much smaller than METIS's large coarsening data structures.

**Integration with Full-Graph Training**: We use METIS when host memory is sufficient. Otherwise, switching-aware partitioning offers a fast and memory-efficient alternative with good partition quality. For comparison with Spinner [63] and SOTA out-of-core partitioner (2PS-L [64]), see Appendix P.

### 4.4 Implementation: `PyGriNNder`

Users with PyG [26] code can utilize GriNNder by simply inheriting the `GriNNderGNN` module. Users only need to implement the `layer_forward` method in addition to the default `forward` method. See Appendix L for the API example, framework structure, and implementation details.

## 5 Evaluation

### 5.1 Experimental settings and baselines

We provide a brief overview of experimental settings and baselines. For more details, see Appendix M.

**Models/datasets**: We use 3-/5-layer GCN [52] with a hidden dimension of 256. We also test GAT [87] and GraphSAGE [33]. Datasets range from medium to large scale: Products [38], IGBM [51], and Papers [38]. We also utilized Kronecker graphs [54] (average degree=10) with the random initial feature of dimension 128 and #classes of ten.

**Hardware**: We run main experiments on a workstation with an AMD Ryzen9 7950X 3D CPU (16C 32T), 128GB DDR5-5600 RAM, one RTX A5000 (24GB) GPU, a PCIe 5.0 NVMe SSD (4TB), and a total 4TB swap space for swap-based experiments. For distributed baselines, we use a 4-server cluster; each node has four RTX A6000 GPUs interconnected by NVLink [69] and Infiniband SDR [68]. For IGBM/Papers, we needed all 16 GPUs to fit the data in the GPU memory. For Products, using fewer GPUs could yield better performance, but we used all GPUs to maintain consistency among datasets.

**Baselines**: *(Training)* We compare GriNNder (GRD) against various single-server/distributed methods: ① Micro-batch training (Betty [97]), ② Micro-batch training with storage extension (Ginex [70]), ③ Host memory offloaded training (HongTu [92]) with OS swap memory, ④ Distributed full-graph training (CAGNET [85]), ⑤ Distributed full-graph training with communication skipping (Sancus [72]), ⑥ Naïve storage extension of full-graph training (ROC [42]). For details of micro-batch and host memory offloaded training, see Appendix B. We showed ⑥ only in Appendix X due to its much slower performance. In the appendix, we also tested two storage-based mini-batch training (⑦ DiskGNN [59], ⑧ GNNDrive [44]) with micro-batch extension (Appendix C). For out-of-memory issues in distributed baselines, we add host-memory checkpointing (*) to enable execution.

GriNNder achieves equal final accuracy with all the baselines (see Appendix W) except ⑤, which is non-exact due to its staleness. All baselines use the state-of-the-art partitioner MT-METIS [53]. For fairness, if METIS exceeds our setting's memory, we assume it was preprocessed elsewhere following the standard practice, except for partitioning experiments. *(Partitioning)* For comparisons with lightweight partitioners, we benchmarked Spinner [63] and an out-of-core partitioner (2PS-L [64]).

## 5.2 Large graph training results

Table 1 presents per-epoch training time for GriNNder (GRD) compared to five base-lines—Betty, Ginex, HongTu, CAGNET, and Sancus—using 3-/5-layer GCNs (hidden dimension 256) on Products, IGBM, and Papers.

**Micro-batch (Betty, Ginex)**: Despite Betty's memory-only design (no storage), GRD achieves up to 30.98× faster training, largely due to Betty's repeated neighbor expansions. Ginex uses storage to extract MFG, yet still suffers from redundant computation caused by neighbor explosion, which GRD improves up to 77.92×.

Table 1: **Results of training time (min)/epoch.**

| | # nodes Method | 2.4M PRODUCTS | 10M IGBM | 100M PAPERS |
|---|---|---|---|---|
| |L|=3 Limited | BETTY | 0.61 | 28.71 | GPU OOM |
| | GINEX | 9.00 | GPU OOM | 17.72 |
| | HONGTU | 0.17 | 6.46 | Swap OOM |
| | **GRD** | **0.12** | **0.93** | **9.07** |
| Dist. | CAGNET | 0.21 | 1.41 | *10.01 |
| | SANCUS | 0.19 | *0.77 | *GPU OOM |
| |L|=5 Limited | BETTY | 1.05 | GPU OOM | GPU OOM |
| | GINEX | 15.10 | GPU OOM | GPU OOM |
| | HONGTU | 0.32 | 14.90 | Swap OOM |
| | **GRD** | **0.23** | **1.52** | **12.03** |
| Dist. | CAGNET | 0.38 | 2.10 | *GPU OOM |
| | SANCUS | 0.36 | *1.41 | *GPU OOM |

SANCUS: Non-exact full-graph (with staleness)

**Products (Medium)**: Since HongTu can fit Products entirely in host memory, one might expect it to outperform storage-based GRD. In practice, HongTu's redundant snapshots slow it down, allowing GRD to beat it by 1.44/1.40× on 3-/5-layer GCNs.

**IGBM (Large)**: Micro-batch methods suffer from GPU OOM on deeper models—Betty/Ginex often cannot handle the neighbor explosion. HongTu must manage large volumes of data in host memory, drastically increasing overhead. In contrast, GRD is 6.97/9.78× faster than HongTu with caching and non-redundancy. Even against multi-GPU CAGNET, GRD achieves faster speed (1.52/1.38×).

**Papers (100M)**: This highlights GRD's scalability. Betty and Ginex often fail on deeper models with OOM from neighbor explosion, and HongTu from activation snapshots. GRD avoids these with efficient caching and no redundant snapshots. Ginex can run the 3-layer model but is 1.95× slower than GRD. Notably, GRD's speed is faster than CAGNET (1.10×) despite using a single GPU.

Table 2: **Training time (min)/epoch sensitivity for graph sizes with synthetic graphs.** For more results with ablation, see Appendix N.

| | # nodes | 4.2M | 8.4M | 16.8M | 33.6M |
|---|---|---|---|---|---|
| |L|=3 | HONGTU | 0.43 | 0.83 | 7.25 | 36.31 |
| | **GRD** | **0.29** | **0.59** | **1.93** | **3.73** |
| |L|=5 | HONGTU | 0.83 | 1.99 | 19.15 | 96.99 |
| | **GRD** | **0.57** | **1.14** | **3.71** | **7.76** |

**Additional–synthetic**: In Table 2, we tested various-sized Kronecker graphs to validate scalability. GriNNder provides stable speedup over HongTu (1.41-12.50×).

## 5.3 Ablation by decreasing effective cache size and cache hit rate

Table 3 analyzes GriNNder 's sensitivity to effective cache size by varying the hidden dimension on IGBM. We ablated GriNNder: HongTu, HongTu + grad-engine activation regathering (GRD-G), and GRD-G + partition-aware graph caching (GRD-GC). GriN-Nder outperforms HongTu by 6.84–12.34×. When host memory can cache most data (in 3 layers), GRD-G alone yields improvements over HongTu. However, in 5-layer settings, host memory becomes a bottle-neck, making cache replacement crucial. Thus, GRD-GC gains 3.09-4.04× speedup over GRD-G. Overall, GriNNder is robust on cache sizes. Also, we reported the cache hit rates in Appendix O. Larger datasets incur more reuse from the higher #partitions, making the hit rates significant (53.70-92.77%).

Table 3: **Sensitivity on effective cache size with ablation (training time (min)/epoch).**

| | # hiddens Method | |H|=384 0.75 $ SIZE | |H|=512 0.5 $ SIZE | |H|=1024 0.25 $ SIZE |
|---|---|---|---|---|
| |L|=3 | HONGTU | 12.53 | 18.67 | 39.32 |
| | GRD-G | **1.20** | **1.51** | 20.68 |
| | GRD-GC | 1.41 | 1.91 | **3.98** |
| |L|=5 | HONGTU | 25.07 | 31.81 | 93.42 |
| | GRD-G | 10.26 | 12.50 | 42.14 |
| | GRD-GC | **2.54** | **3.37** | **13.65** |

## 5.4 Analysis on host memory usage

Figure 7a shows an ablation study on how GriNNder reduces host memory consumption. We compare GRD-G (i.e., HongTu + grad-engine activation regathering) and GRD-GC (GRD-G + partition-aware graph caching). HongTu suffers from snapshots, while GRD-G eliminates them. GRD-GC's layer-wise up/offload further cuts the peak usage from HongTu by 5.75×. Figure 7b shows the host memory usage timeline. With GDS and caching, GRD-GC shows significantly low memory usage.

**GRD-G** : +Grad-Engine Activation Regathering
**GRD-GC**: GRD-G + Cached Storage Offloading

Figure 7: Host memory usage of GriNNder on the IGBM.

## 5.5 Analysis on partitioning algorithms

Table 4: **Memory usage (GB) of partitioning.**

| Dataset | Method | Graph | Part. | Label | Add. | Total |
|---|---|---|---|---|---|---|
| PRODUCTS | MT-METIS | 1.01 | 0.01 | | 9.93 | 10.95 |
| | **GRD** | 1.01 | 0.01 | | **0.52** | **1.54** |
| IGBM | MT-METIS | 1.12 | 0.04 | | 28.34 | 29.50 |
| | **GRD** | 1.12 | 0.04 | | **0.87** | **2.03** |
| PAPERS | MT-METIS | 26.71 | 0.44 | | 867.84 | 895.00 |
| | **GRD** | 26.71 | 0.44 | | **9.56** | **36.72** |

Table 5: **Effect of partitioning (left) and model type (right) on training time/epoch (sec).**

| Method | PRODUCTS | IGBM |
|---|---|---|
| MT-METIS | 6.93 | 55.62 |
| Random | 14.73 | 353.06 |
| GRD | 9.26 | 125.87 |

| Model | $\|L\|$ | HongTu | GRD |
|---|---|---|---|
| GAT | 3 | 741.07 | **65.52** |
| | 5 | 1153.82 | **108.01** |
| SAGE | 3 | 584.76 | **69.24** |
| | 5 | 794.13 | **112.70** |

**Memory usage**: Table 4 shows that GriNNder 's partitioning greatly reduces memory usage by 7.10–24.37× compared to METIS. METIS requires additional memory for coarsening-phase intermediates. In contrast, switching-aware partitioning only needs $\mathcal{O}(|E|)$ extra space.

**Convergence/practical overhead/comparison with other partitioners**: We also reported the trend of partitioning quality improvement (convergence) and practical overhead of switching-aware partitioning in Appendix Q. We observed that at most 50 iterations are enough for convergence. Also, the practical overhead of partitioning in the actual training was only 0.07/0.02/0.39% of the total training time on Products/IGBM/Papers, respectively. We additionally benchmarked the (time-to) quality of GriNNder compared to lightweight partitioners (Spinner and 2PS-L) (see Appendix P). Switching-aware partitioning quickly results in higher-quality partitions for both cases.

**Partition/training time**: Among datasets, only Papers exceeded the host memory capacity. Partitioning it into 16 parts with METIS triggers host swap due to its large memory demand and took 77.26 (min), making switching-aware partitioning 10.51× faster (7.35 (min)). Table 5 (left) evaluates how partitions affect the training of 3-layer GCNs on Products and IGBM. Although METIS—with near-optimal partitions—yields the shortest training time, it uses significantly more memory. Switching-aware partitioning needs far less memory while improving training speed by 1.59× on Products and 2.80× on IGBM over random partitioning.

## 5.6 Other sensitivity studies (model, configuration, and multi-GPU sensitivity)

Table 5 (right) shows GriNNder with GAT [87] and GraphSAGE [33], using IGBM. GriNNder maintains 7.05–11.31× speedup over HongTu, demonstrating its efficiency beyond GCN. We examine the impact of #partitions configuration on the 3-layer setting in Appendix R. Compared to HongTu, from the efficient caching and redundancy elimination, GriNNder is much more robust on the configuration. We also tested the multi-GPU scalability in Appendix S. Although GriNNder was not designed for multi-GPU environments, it is scalable to some degree (up to 2.44× with four GPUs).

## 6 Conclusion

To our knowledge, GriNNder is the first work on full-graph GNN training with storage offloading in limited resources. Its careful optimizations enable full-graph training of large datasets previously impossible in conventional frameworks. We will open-source GriNNder to facilitate its use.

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

## A   Survey of 2024 conferences' submission on GNN domains

In our survey of NeurIPS/ICML/ICLR 2024 papers, a total of 76 papers are related to GNN domains. In 76 papers, 44.7% (34 papers) used full-graph training, and among them, 38.2% (13 papers) directly reported out-of-memory. In terms of experimental environments, a total of 62 papers reported their GPU environments, and 45 papers utilized a single GPU (72.6%). Also, some papers with full-graph training directly stated that larger-sized datasets can incur out-of-memory when running their experiments. This shows the importance of enabling full-graph training of large graphs under limited resources (e.g., a single GPU).

## B   Drawbacks of existing full-graph training methods for limited environments

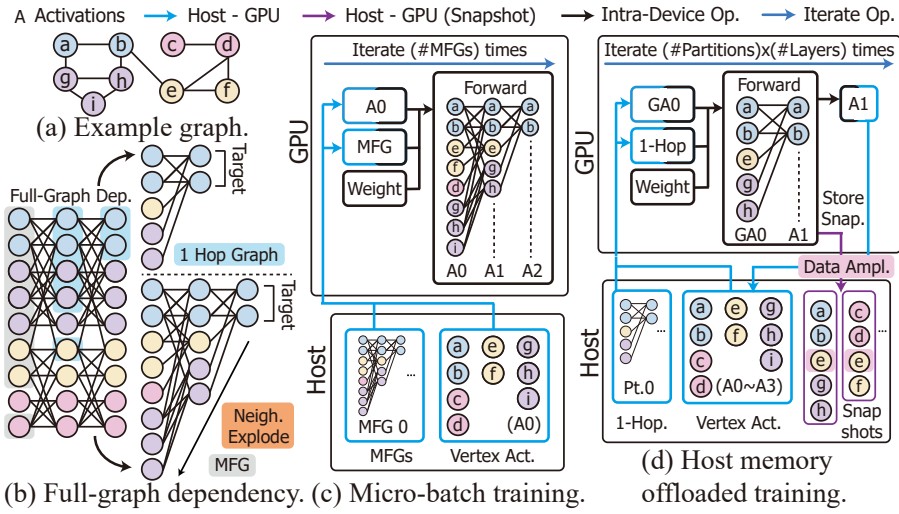

Figure 8: Full-graph training of single epoch for limited resources.

Full-graph GNN training processes all vertices' activations and gradients in a single pass, requiring substantial memory. A few existing approaches exist for single-server approaches. Micro-batch [97] and host memory offloaded [92] training have tried to conduct full-graph training in GPU memory-limited environments. Figure 8 illustrates an example graph and discusses the drawbacks of the above methods based on the full-graph dependency.

**Micro-batch training**: Betty [97] (Figure 8c) accumulates gradients from message flow graphs (MFGs) with all neighbor information across all layers, followed by a single weight update. However, even a small number GNN layers cause MFGs to expand rapidly (Figure 8b), often exceeding the GPU memory. Partitioning [47, 49, 53] can reduce MFG size but requires significant memory, presenting a practical bottleneck.

**Host memory offloaded training**: HongTu [92] (Figure 8d) reduces GPU memory usage by moving activations and gradients to host memory. A 1-hop partitioning approach extracts 1-hop graphs that fit in GPU memory. During an epoch, activations for these graphs are transferred to the GPU, processed, and offloaded back to host memory. While this saves GPU resources, it causes a *data amplification* problem: by saving 'snapshots' of 1-hop graphs for the backward pass, vertices appearing in multiple 1-hop graphs are stored repeatedly, increasing memory and I/O overhead.

**Impractical partitioning**: Both micro-batch [97] and host memory offloaded [92] methods rely on partitioning tools like METIS [47, 49, 53], which sequentially coarsen and refine graphs. This process consumes up to $4.8\times$ the graph's size in memory [50], often exceeding the capacity of a typical single server. Hence, existing single-server full-graph methods face out-of-memory risks or resort to an external server.

## C   Limitation of extending storage-based mini-batch training to full-graph training with micro-batch training

While extending storage-based mini-batch training (e.g., Ginex [70], MariusGNN [88], DiskGNN [59], GNNDrive [44]) to full-graph training by setting the batch size to the entire node set and maximizing the neighbor size (i.e., micro-batch training from Betty [97]) may seem to enable the large-scale full-graph training on a limited environment, it faces several limitations.

First, as it still depends on the message flow graph structure (MFG), it faces the GPU memory limit like the original micro-batch training (Betty). Since micro-batch training needs to keep all neighbor information intact without sampling, it easily falls into the out-of-memory due to neighbor explosion. For more details, please refer to Appendix B.

Second, they are mainly focused on handling initial features efficiently for mini-batch training, and are inefficient in supporting full-graph training without sampling. For instance, DiskGNN aggressively utilizes the preprocessing and pre-stores the mini-batch message flow graphs and related initial features to efficiently support large-scale mini-batch training with storage. However, since full-graph training (with micro-batch training) needs to handle all the features without dropping, the preprocessed data size easily exceeds the SSD capacity due to the redundant saved data.

Table 6: **Performance of extending storage-based mini-batch training to full-graph training.** [†]: GNNDrive's GPU caching is statically preprocessed, so the fanout is restricted to 25 and not equivalent to full-graph training. [*]: Preprocessing failed because of excessive disk space usage.

| Method | Products | IGBM | Papers |
|---|---|---|---|
| Ginex | 9.00 | OOM | 17.72 |
| DiskGNN | 2.18 | Preproc. Fail[*] | Preproc. Fail[*] |
| GNNDrive | 6.33[†] | OOM | 12.06[†] |
| **GriNNder (Ours)** | **0.12** | **0.93** | **9.07** |

To show the above limitations directly, we evaluated DiskGNN [59] and GNNDrive [44], which surpass the previous state-of-the-art storage-based mini-batch training (Ginex [70], MariusGNN [88]) in Table 6. Ginex and GNNDrive encountered GPU out-of-memory on IGBM due to neighbor explosion without information dropping. On Papers, even with fanout 25, they were significantly slower than ours. DiskGNN uses offline preprocessing to pre-store all cacheable mini-batches with features. The preprocessing of IGBM/Papers fails by overflowing 4TB SSD, even with reduced fanout (25) from neighbor explosion. Our method is significantly faster for runnable cases (Products/Papers).

To sum up, while mini-batch storage-based systems can emulate full-graph training by micro-batch training, results show that this becomes infeasible on a large scale. This is due to either GPU memory exhaustion or prohibitive preprocessing disk usage. GriNNder avoids them by not relying on message flow graphs (MFGs) or redundant preprocessing.

## D   Overall procedure of GriNNder

As GriNNder is the first work on storage offloaded full-graph GNN training, we carefully designed the framework to address the three challenges outlined in Section 1, whose overall procedure is listed in Algorithm 1. GriNNder first partitions the graph into smaller pieces, which should be done to incur minimal data transfer (line 2). Our contribution is on devising a lightweight partitioning algorithm that operates with significantly lower memory requirements while preserving the partitioning quality (Section 4.3). Then, for each partition (lines 10 and 21), forward and backward passes are performed on the GPUs (lines 11-14 and 22-27). To maximize the reuse of the data, GriNNder designs an efficient policy to cache intermediate data on the host memory (Section 4.1). During the forward/backward passes, much of the data transfer occurs between GPU-CPU due to checkpointing (lines 12, 14, 24, 26). This was originally designed toward reducing latency in previous work[92], but it severely increases the amount of traffic and host memory usage for storage offloading scenarios. GriNNder redesigns the gradient engine with redundancy elimination, achieving significantly higher speedup and less memory requirement (Section 4.2).

**Algorithm 1** Overall procedure of GriNNder

**Input:** $\{W^i | 1 \leq i \leq L\}$: initial parameters, $L$: #layers
   $G$: graph, $F$: initial features, $P$: #partitions to meet GPU mem. req.
**Output:** $\{W^i | 1 \leq i \leq L\}$: updated parameters
**Notations:**
   $T_p$: 1-hop topologies (src→dst)
   $A_p^l$: destination features/activations of layer $l$, partition $T_p$
   $GA_p^l$: gathered source features/activations of layer $l$, partition $T_p$
1: **if** $METIS_{limit} \geq Host_{limit}$ **then**
2:   $T_{(\cdot)} \leftarrow SA\_Partition(G, P)$   // Switching-aware partitioning (Sec. 4.3)
3: **else** $T_{(\cdot)} \leftarrow METIS(G, P)$ **end if**
4:   // Do until finding proper $P$ which makes all $T_p$s fit GPU memory limit.
5:
6: **for** $e = 1 \dots \#epochs$ **do**
7:   // Forward pass
8:   **for** $l = 1 \dots L$ **do**
9:     $Storage\_to\_Host(A_{(\cdot)}^{l-1})$ // Partition-aware graph caching (Section 4.1)
10:    **for** $p = 0 \dots P - 1$ **do**
11:      $GA_p^{l-1} \leftarrow Gather(A_{(\cdot)}^{l-1})$
12:      $Host\_to\_GPU(GA_p^{l-1})$
13:      $A_p^l \leftarrow FW(W^l, GA_p^{l-1}, T_p)$   // w/ Regathering (redundancy elimination) (Section 4.2)

14:      $GPU\_to\_Host(A_p^l)$
15:    **end for**
16:   **end for**
17:   // Backward pass
18:   **for** $l = L \dots 2$ **do**
19:     // Partition-aware graph caching (Section 4.1)
20:      $Host\_Upload\_or\_Intitialization(A_{(\cdot)}^{l-1}, \nabla A_{(\cdot)}^{l-1})$   // Host as write-back buffer
21:      **for** $p = 0 \dots P - 1$ **do**
22:        $Storage\_to\_Host(A_p^l, \nabla A_p^l)$
23:        $GA_p^{l-1} \leftarrow Gather(A_{(\cdot)}^{l-1})$
24:        $Host\_to\_GPU(GA_p^{l-1})$      // Grad-engine activation regathering (Section 4.2)
25:        $(\nabla GA_p^{l-1}, \nabla W^l) \xleftarrow{\cdot, +} BW(W^l, A_p^l, \nabla A_p^l, GA_p^{l-1})$
26:        $GPU\_to\_Host(GA_p^{l-1})$
27:        $\nabla A_{(\cdot)}^{l-1} \xleftarrow{+} Scatter(GA_p^{l-1})$
28:      **end for**
29: **end for**

## E  Profile of dependency among partitions

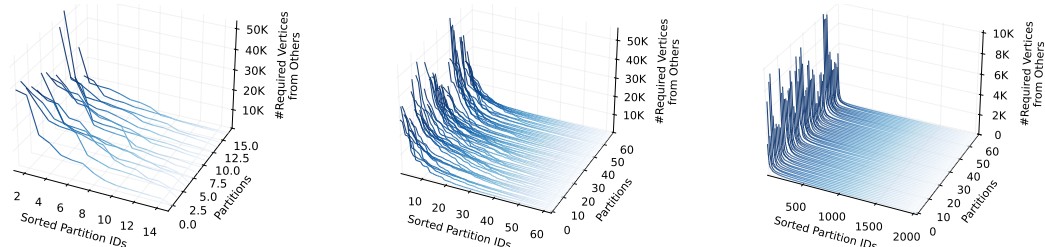

Figure 9: Partition dependency profile. (left) Products with 16 partitions, (mid) IGBM with 64 partitions, and (right) Papers with 2048 partitions. In the case of Papers, we only presented earlier 64 partitions for visibility.

We additionally presented the profile of dependency among partitions on other datasets in Figure 9. When the size of a graph becomes larger, we need to partition the graph into a much larger number of partitions. This makes the trend of power-law distribution clearer. For instance, in Figure 9(right), the Papers dataset with 2048 partitions shows a very vivid power-law distribution compared to the other two cases. This further enhances the scalability of GriNNder on large-scale graphs.

## F  Vertex-wise cache management vs. partition-wise cache management

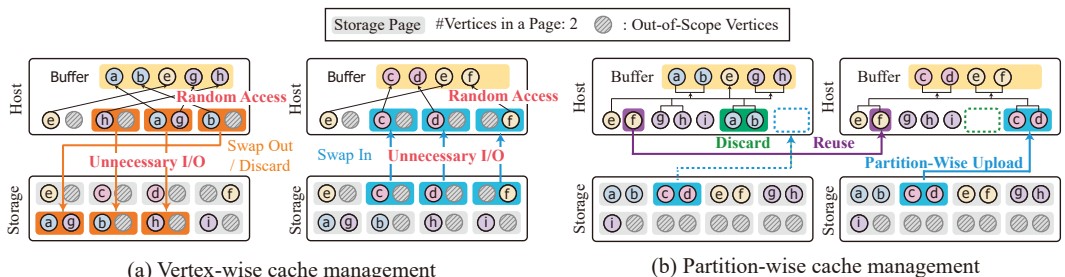

Figure 10: Advantage of partition-wise cache management compared to vertex-wise one.

Figure 10 emphasizes the advantage of partition-wise cache management compared to the vertex-wise cache management. Since storage devices access data at a page granularity (e.g., 4KB), vertex-wise cache management incurs a substantial amount of *unnecessary I/O*, denoted as 'Out-of-Scope' vertices. For instance, when processing the next partition, in Figure 10a, vertex-wise management needs to swap out (or discard) and swap in unnecessary data combined with the required data due to the page granularity of a storage device. In contrast, loading and evicting at a partition granularity alleviates such overhead.

## G  I/O optimizations of GriNNder

### G.1  Overlapping of processing and cache management

GriNNder schedules host memory cache evictions and prefetching to overlap with GPU computations, minimizing storage I/O latency as illustrated in Figure 11. ① We pick the next target partition to exploit already-cached neighbors, determined statically since 1-hop graphs are fixed. ② We discard partitions no longer needed. ③ We fetch only required partitions from storage while keeping reusable ones in the host memory. Because we keep a small extra buffer (dotted blue), uploading the dependency for partition 1 (pre-compute) does not have to wait for partition 0 computation and the succeeding evictions (post-compute), enables overlapping these I/O operations (②') with ongoing computations. Also, we overlap the GPU compute and host–GPU I/O to further reduce latency.

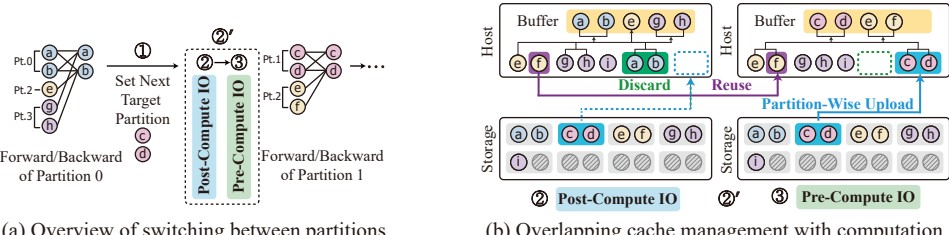

(a) Overview of switching between partitions    (b) Overlapping cache management with computation

Figure 11: Overview of overlapping cache management with computation.

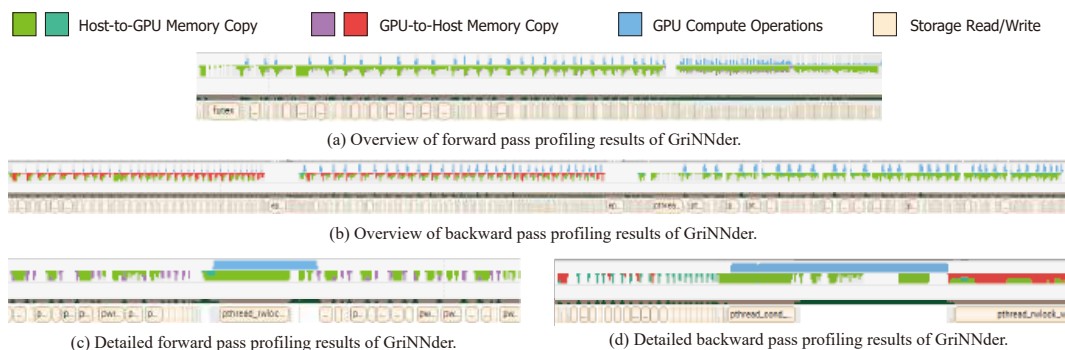

(a) Overview of forward pass profiling results of GriNNder.

(b) Overview of backward pass profiling results of GriNNder.

(c) Detailed forward pass profiling results of GriNNder.    (d) Detailed backward pass profiling results of GriNNder.

Figure 12: Profiling results of GriNNder's forward and backward pass.

We actually profile the training procedure of GriNNder, as illustrated in Figure 12. We profiled the 3-layer GCN on the IGBM dataset with #partitions=32. In both forward and backward passes, GriNNder overlaps the host memory and storage I/O with the GPU computation. Thus, in overall training, GriNNder enables aggressive latency overlapping of I/O and computation and provides superior training throughput.

## G.2 In-partition vertex ordering for sequential accesses

Another source of slowdown is in the gathering, which places vertex activation to be sent ($GA$) to the GPU in a dedicated host buffer. This involves multiple random memory access, as illustrated in Figure 10a, causing slowdown. To avoid this, after the graph is partitioned, we reorder the individual adjacency lists such that the neighbors are first sorted by their partition IDs and then by their vertex IDs. This replaces the random lookups with a single random lookup per partition, as in Figure 10b.

# H Comparison with HongTu's gradient engine

HongTu [92] (Figure 13a) mitigates the PyTorch autograd's issue by recomputing intermediate activations on demand. It designed a gradient engine to snapshot the gathered activations toward reducing latency (through enabling sequential accesses to snapshots), but at the cost of increased snapshots and redundant vertex data across partitions. As a result, each vertex may be stored up to $\alpha$ times (the average expansion ratio of 1-hop graphs), which adversely impacts memory consumption and bandwidth requirements, particularly for large datasets. This is because it assumes abundant host memory and does not consider the employment of storage, which has much lower bandwidth compared to the host memory.

Figure 13b illustrates the proposed grad-engine activation regathering, which skips snapshot creation during the forward pass. Instead, whenever the backward pass requires input activations ($GA$), we *regather* them just-in-time from the *already-stored* activations ($A$) managed by partition-aware graph caching (see Section 4.1) in a partition-wise manner. This replaces repeated snapshot storage with lightweight data arrangement, significantly reducing host memory usage and I/O volume while maintaining algorithm correctness. While HongTu does suggest an additional strategy of storing the aggregated intermediate values (*IO*) instead of the activation values (*AO*), this is only applicable to

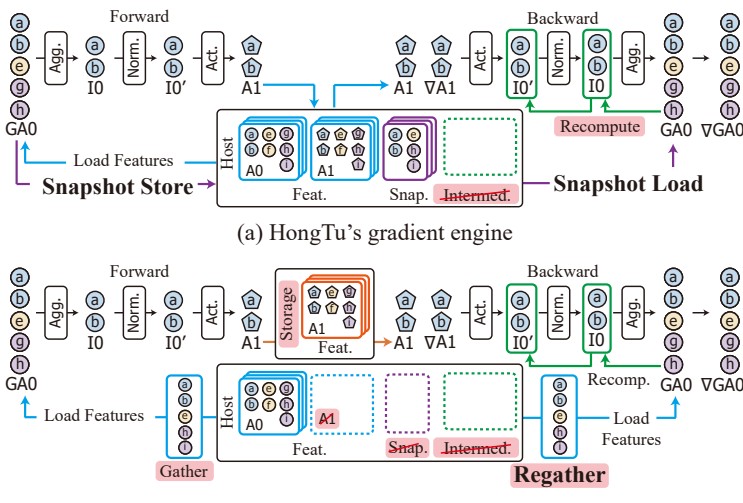

(a) HongTu's gradient engine

(b) Grad-engine activation regathering (Ours)

Figure 13: Comparison with HongTu [92], which does not consider the employment of storage.

GCN-style models. Contrarily, grad-engine activation regathering generally applies to any model structure (e.g., GAT).

# I   In-depth I/O volume and memory footprint analyses

Table 7: **I/O analysis in forward pass**

| Methods | GPU-Host | Host-Storage | GPU-Storage |
|---|---|---|---|
| HongTu w/ OS swap memory (i.e., `mmap`) | $(2\alpha + 1)D$ | $(2\alpha + 1)D - Mem_{Host}$ | |
| Ours | $\alpha D$ | $\alpha D - CacheHit$ | $D$ |

\* $|V||H| = D$. Topology data I/O is omitted for brevity.

Table 8: **Maximum memory usage analysis**

| Methods | Host | Storage |
|---|---|---|
| HongTu | $(\alpha + 1)D|L| + 2D$ | |
| Ours | $D + D$ | $D|L| + D$ |

\* $|V||H| = D$. Considers activation and gradients.

Grad-engine activation regathering greatly reduces the I/O volume from snapshot store/load per layer and the memory footprint displayed in Table 7 and Table 8, where $D = |V||H|$.

**I/O Volume**: We assume host memory offloaded training (HongTu [92], see Appendix H for the detailed I/O procedure) to utilize OS swap memory (i.e., `mmap`), since it targets the host memory, not storage employment. In Table 7, compared to HongTu, the input activation-related GPU-host I/O volume ($2\alpha D$) is halved ($\alpha D$) by skipping snapshots. GriNNder incurs $\alpha D - CacheHit$ amount of host-storage traffic for intra-layer partition-wise caching, but this is significantly less than utilizing `mmap` swap memory. Also, when host memory can handle the single-layer activations ($D$), this term becomes $D$ from a full hit. When the host memory offloaded training faces the memory limit ($Mem_{Host}$), it needs to swap around $(2\alpha + 1)D - Mem_{Host}$ data from/to storage. Given that $\alpha$ is around 3-10, the improvement is significant.

**Memory Footprint**: In Table 8, we report the peak memory usage of host offloaded training [92] (HongTu) and GriNNder. For HongTu, the overhead mostly comes from storing snapshots for all layers. These redundant snapshots consume additional $\alpha D|L|$ on top of $D|L|$ activations. It needs to save $2D$ of gradients in backward pass to handle input and output gradients. In contrast, with

828 grad-engine activation regathering and partition-aware graph caching, GriNNder consumes up to
829 $D + D$ host memory for saving layer-wise activations and gradients. Regarding storage usage,
830 GriNNder consumes $D|L|$ for saving activations and $D$ for single-layer gradients.

# J   METIS and its memory usage

832 Sequential partitioning (e.g., METIS) comprises three stages: coarsening, initial partitioning, and
833 un-coarsening. In the coarsening phase, it tries to generate a good initial partitioning, which can
834 be partitioned to the initial partitioning state. From this state, the un-coarsening phase refines the
835 boundaries of partitions to produce better partitioning results. This complex procedure incurs huge
836 memory requirements when using large graphs [50]. To save coarsened intermediate graphs, sequatial
837 partitioning requires $O(2|V| + |E| + \sum_{i=1}^{L} |E_i| + |V_i|)$ memory where $|(\cdot)_i|$ is for coarsened graphs
838 and $L$ is the number of levels of coarsening. [50] reported that it consumes at least $4.8\times$ more
839 memory than the graph data itself ($|V| + |E|$). As a result, this huge memory consumption harms the
840 practicability of existing full-graph training for limited resources [97, 92].

# K   Insights and details of switching-aware partitioning

842 Existing partitioners (e.g., METIS-based [47, 49, 53, 91, 60]) output near-optimal partitions but
843 often exceed single-server memory limits (Figure 6a). This makes prior approaches impractical, as
844 partitioning needs to be iterated to find the adequate number of partitions to fit in the GPU. While
845 offline partitioning is possible, each new environment demands re-partitioning, making a lightweight
846 partitioning method essential.

847 We draw inspiration from streaming partitioning (Spinner [63]), which applied traditional label
848 propagation [108] to partitioning in distributed cloud graph systems (e.g., Pregel [62]). While
849 lightweight label propagation suits our host memory constraints, Spinner's message-passing-based
850 design is unsuitable for such limited environments.

851 Hence, we propose switching-aware partitioning, which adapts label propagation for limited resources
852 with memory usage similar to CSR. We also introduce a group-wise propagation strategy suited for
853 storage-offloaded full-graph training.

854 Switching-aware partitioning aims to find vertices with similar properties in different partitions and
855 relocate them to the same partition. Additionally, we need to balance the size of each partition to
856 reduce the workload imbalance between partitions. To do so, we iteratively refine the partitions by
857 selectively relocating vertices within a certain limit.

858 Figure 14 shows the detailed procedure of the proposed switching-aware partitioning. At first, we
859 set the initial partitioning state ($S_0 = P_0, ..., P_{p-1}$) by randomly assigning each vertex to different
860 partitions. We want to achieve high-quality partitioning while maintaining the number of vertices
861 of all partitions close to $|V|/p$. $|\cdot|$ means the #vertices in a partition (or a graph), and $p$ is the
862 #partitions. We additionally define the maximum capacity term as $\beta$ and set the maximum capacity
863 limit of a single partition as $\beta \times |V|/p$. Here the capacity of a partition refers to the number of
864 vertices allocated to said partition. In a state $S_i$, each partition $j$ has the available relocation capacity
865 ($RC_{(i,j)}$) as follows:

$$RC_{(i,j)} = \beta \times |V|/p - |P_j|, (0 \le j < p) \tag{1}$$

866 This is used to limit the number of vertices moved to the current partition. Figure 14a illustrates the
867 intermediate state ($S_i$) where each partition has the available relocation capacity of six ($RC_{(i,j)} = 6$).
868 Following the CSR format, our data structure comprises source pointers (`SrcPtr`) and destination
869 indices (`DstIdx`). We manage another array (`Dst's Partition`) and fill this array with the partition
870 of each destination index in `DstIdx`. For example in Figure 14a, the vertex 0 has neighbors of vertex
871 $\{1, 2, 5, 7, 4, 3\}$. For each neighbor, we fill the `Dst's Partition` with its partition $\{2, 2, 2, 0, 1, 1\}$.

872 From a state ($S_i$) (Figure 14a), we calculate *kth preference* of a vertex: among the neighbors of the
873 vertex, the partition ID of the $k$th largest frequency is the $k$th preference of the vertex. Then, using
874 each vertex's first preference, each partition manages its own relocation candidate vertices from other
875 partitions. In Figure 14b, vertex 0's neighboring vertices' partitions are $\{2, 2, 2, 0, 1, 1\}$. Among
876 them, the partition that occurs most frequently is 2. Therefore, we put vertex 0 to the partition 2's
877 relocation candidate (0 is now included in Pt.2 List in Figure 14b).

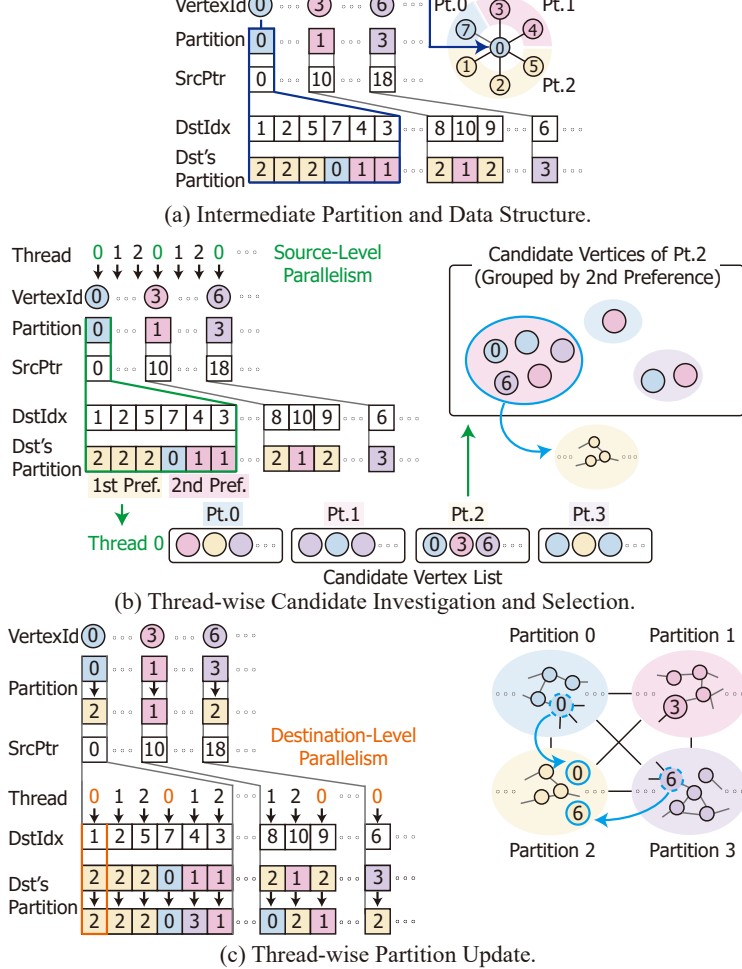

(a) Intermediate Partition and Data Structure.

(b) Thread-wise Candidate Investigation and Selection.

(c) Thread-wise Partition Update.

Figure 14: Switching-aware partitioning.

When selecting the final vertices to be relocated among candidates, we select them in a group-wise manner. In Figure 14b, we first use the 2nd preference partition of each vertex as a feature to help cluster vertices into different groups, unlike the baseline streaming partitioning algorithm. We then choose the largest group with the same 2nd preference to avoid vertices belonging to small, disparate clusters being relocated. In this example, vertex 0's 2nd preference partition is partition 1, and vertex 6's 2nd preference partition is also partition 1. Therefore, we put those two vertices into the same group. We choose to relocate the group including $\{0, 6\}$ because it is the largest group among the candidates. This provides a clustering effect and helps the convergence speed of partitioning. This can be generalized into comparing until $k$th preference, but we use $k = 2$ as default because it already empirically provides good performance.

To parallelize the procedure, we apply source-level parallelism, which distributes the source vertices to each thread. Each thread manages its own candidate lists for partitions as depicted in Figure 14b with the example of thread 0. We dedicate each thread to the equal available relocation capacity $(RC_{(i,j)}/\#threads)$ to run threads in a fully parallel manner.

After selection, we update the relocation result to the `Dst's Partition` array. In Figure 14c, vertex 0 and 6 are selected to be relocated to partition 2. Therefore, we update the values of vertex 0 and 6 in `Dst's Partition` array to 2 (meaning partition 2). This procedure is conducted with destination-level parallel, as illustrated in Figure 14c. After the update, using the updated data structure, iteration $i + 1$ proceeds. For each iteration $i$, we conduct the following procedure until reaching the termination condition, which will be discussed in the next subsection.

## K.1 Detailed terms and memory efficiency

We discussed switching-aware partitioning as a procedural view. In the detailed algorithm, we need a penalty term for suppressing the propagation to reduce the imbalance among the number of vertices in partitions. Therefore, in a state $S_i$, for a vertex $v$, the scoring term ($Score_{(v,I,j)}$) for each partition $j$ and the final objective are as follows:

$$Penalty_{(i,j)} = |P_j|/(\alpha \times |V|/p), (0 \leq j < p)$$
$$Score_{(v,i,j)} = 1 + \#N(v,j)/\#N(v,\cdot) - Penalty_{(i,j)} \tag{2}$$
$$maximize \sum_{v \in G} Score_{(v,i,j=partition_v)}$$

where $\#N(v,j)$ denotes the frequency of partition $j$ among the neighbors of the vertex $v$ and $Penalty_{(i,j)}$ denotes the penalty term of state $S_i$ of partition $j$. The penalty term reduces the preference when a partition already reaches the additional capacity $\alpha$. The objective function calculates the total sum of the internal preferential scores of partitions. The partitioning halts when the objective does not improve over $\epsilon = 0.001$ for $w = 5$ times.

In terms of memory consumption, switching-aware partitioning requires a significantly small amount of memory. As we only utilize `SrcPtr`, `DstIdx` from CSR, `Dst's Partition` and the partition label, switching-aware partitioning only consumes $O(2|V| + 2|E|)$ amount memory. This is significantly less memory usage than METIS, which requires huge memory to save intermediate coarsening information.

## K.2 Partitioning in actual training

Sequential partitioning methods [47, 49, 53] provide a near-optimal partitioning while consuming large memory. On the other hand, switching-aware partitioning, provides efficient memory usage while maintaining reasonable partitioning quality. Therefore, when the host memory size is enough to handle partitioning with a sequential partitioning algorithm (i.e., METIS), we fall back to partitioning with it.

# L   API example, framework structure, and implementation

```
from torch_geometric.nn import GCNConv
from torch_sparse import SparseTensor
from models import GriNNderGNN
from utils.loader import GriNNderLoader

class GCN(GriNNderGNN):
    def __init__(..., loader: GriNNderLoader, ... ,
                    use_cache: bool, storage_offload: bool, ...):
        super().__init__(... , loader, ..., use_cache, storage_offload, ...)

        for i in range(num_layers):
            conv = GCNConv(in_dim, out_dim)
            self.convs.append(conv)

    def forward(self, x: Tensor, adj: SparseTensor, ...):
        for (conv, ...) in zip(self.convs[:-1], ...):
            h = conv(x, adj)

    def forward_layer(self, layer, x: Tensor, adj: SparseTensor, ...):
        h = self.convs[layer](x, adj)
```

```
┌─────────────────────────────────────┐
│ ■ Inheriting GriNNderGNN             │
│ ■ GriNNderLoader for storage         │
│ ■ Additional definition              │
│   of forward_layer                   │
└─────────────────────────────────────┘
```

Figure 15: User interface of GriNNder.

Figure 15 shows the user interface of GriNNder. If a user has a model code for PyG [26], the user can utilize GriNNder by simply inheriting the `GriNNderGNN` module and implementing `layer_forward` method. As offloaded full-graph training is layer-wise, a user needs to implement the `layer_forward` method in addition to the default `forward` method of a PyG model.

Figure 16 illustrates the overall framework structure of GriNNder. In **User-Level**, GriNNder provides the base GNN module for inheritance and custom dataloader, which serves offloading-related data and interacts with switching-aware partitioning. In **Middleware**, the offloading engine of GriNNder controls the AIO engine for host-storage I/O and the GPUDirect Storage (GDS) for GPU-storage I/O. The offloading engine also provides the features of GriNNder internally. GriNNder engine utilizes

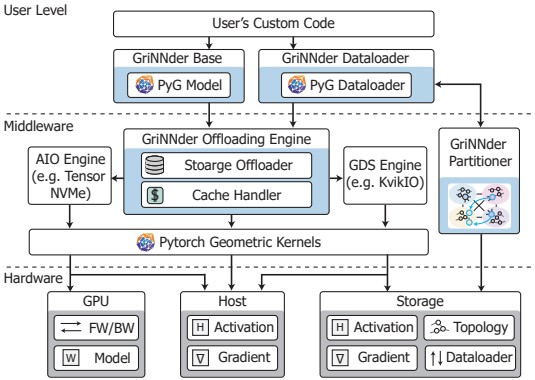

Figure 16: Framework structure of GriNNder.

PyG kernels and implementations for the forward/backward passes. The middleware operates three **Hardwares**, GPU, host, and storage. As a result, a user can enjoy the full features of GriNNder only by providing train code by inheriting GriNNder module.

We implement GriNNder over the `torch.nn.Module` of PyTorch [71] to enable a user to use GriNNder only by inheriting `GriNNderGNN` module. For host-to-storage I/O, we utilized the AIO interface of Linux wrapped by TensorNVMe [2]. For GPUDirect Storage (GDS) [67], we used Kvikio [3], which is the user interface for GDS. Both I/O engines are managed by a thread pool to trigger asynchronous I/O. For the data loader and partitioner, we implemented them with C++ and served these codes to PyTorch through pybind11 [4]. Inside the `GriNNderGNN` module, `offloading engine` conducts the core functionality of GriNNder by interacting with the AIO engine (e.g., TensorNVMe) and the GDS engine (e.g., Kvikio). Our custom partitioning extension provides the partitioning information to the data loader of GriNNder. Additionally, since offloaded training is usually I/O bound, we further optimize GriNNder using I/O overlapping. Using the bidirectional interconnect (i.e., PCIe), we can overlap offloading the activation/gradients from a partition and uploading the required activation/gradients for the next partition.

## M  Detailed experimental settings and baselines

Table 9: **Real-world graph datasets and hyper-parameters**

| Name | Dataset Info. | | | Hyper-parameter | | |
|---|---|---|---|---|---|---|
| | #Nodes | #Edges | Feat. size | lr | Dropout | #Epochs |
| Products [38] | 2,449K | 61.9M | 100 | 0.003 | 0.3 | 500 |
| IGBM [51] | 10,000K | 120.1M | 1024 | 0.01 | 0.5 | 500 |
| Papers [38] | 111,000K | 1,600M | 128 | 0.01 | 0.5 | 500 |

**Models and datasets.** We tested graph convolutional network (GCN) as the baseline GNN architecture and also used GAT [87] and GraphSAGE [33]. We set the hidden size as the widely-used 256, if not stated otherwise. We used three medium- (Products [38]) to large-scale (IGBM [51] and Papers [38]) datasets (details in Table 9). Products is a co-purchasing network where vertices represent Amazon products and edges indicate products purchased together. IGBM and Papers are citation networks with vertices and edges representing research papers and citations, respectively. We also utilized Kronecker random graphs [54] (average degree=10) with the random initial feature of dimension 128 and #classes of 10 for scalability and versatility test with ablation.

**Hardwares.** We used a single PC with AMD Ryzen9 7950X 3D CPU, 128GB DDR5-5600 host memory, and an RTX A5000 24GB GPU. We equipped a PCIe 5.0 4TB NVMe SSD for the swap memory and GPUDirect Storage (GDS) [67] and AIO [1]. We also set swap memory of 4TB for the swap memory-based evaluations. We used a four-server cluster to test distributed full-graph training baselines, each server having four RTX A6000 GPUs, which aggregates to 16 GPUs. Intra-server GPUs are connected via NVLink Bridge [69], and servers are connected via Infiniband SDR [68]. Each server has 512GB DDR4 RAM and an EPYC 7302 (16C 32T). For IGBM/Papers, we needed

all 16 GPUs to fit the data in the GPU memory. For Products, using fewer GPUs could yield better performance, but we used all GPUs to maintain consistency among datasets.

**Baselines.** We compared four single-server baselines with GriNNder (denoted as 'GRD'). For MFG-based full-graph training, we used Betty [97] (called micro-batch training), the state-of-the-art full-graph training in limited environments, as our baseline. As Betty sometimes shows significant slowdowns due to slow MFG generation, we excluded the MFG generation time for comparison. To test extension of storage-based mini-batch training to full-graph training while utilizing SSD, we extend Ginex [70] to micro-batch training [97]. For host offloaded full-graph training, we faithfully implemented HongTu [92] and used it as a baseline. When the training data overflows the host memory, we use storage swap memory to compare it with GriNNder regarding storage usage. We also tested the naïve extension of ROC [42] to naïvely just use storage for offloading, but reported the results of it only in Appendix X because this extension was much slower than the others. In the appendix, we additionally tested two storage-based mini-batch training (DiskGNN [59] and GNNDrive [44]) with micro-batch extension (Appendix C).

We also compared GriNNder with two distributed full-graph training baselines, CAGNET [85] and Sancus [72]. CAGNET is one of the famous distributed full-graph training methods, and Sancus accelerated it by storing stale activations and gradients to reduce the communication bottleneck. Note that while Sancus is not the exact full-graph training from using stale activations and gradients, we still included it as it is one of the state-of-the-art distributed full-graph training frameworks. These two baselines ran on the cluster mentioned above. When GPU out-of-memory issues arise in distributed training baselines, we implement host memory activation checkpointing (indicated by '*') to attempt to make them executable.

For partitioning, we utilized the multi-threaded METIS (MT-METIS) [53] as the baseline, which is one of the state-of-the-art METIS parallelizations (denoted as 'METIS'). Even when it does not run on the testbed due to insufficient memory, we assume it was preprocessed in another environment since all baseline methods rely on METIS. For comparisons with lightweight partitioners, we benchmarked Spinner [63] and an out-of-core partitioner (2PS-L [64]).

# N  Comprehensive analysis with synthetic graph on scalability, ablation, and configuration

We conducted a comprehensive analysis using synthetic graphs, as summarized in Table 10. The tests utilized Kronecker synthetic graphs [54] with sizes ranging from $2^{22}$ to $2^{25}$ nodes (4.2-33.6M) and an average degree of 10.

Across all combinations of layers and datasets, all ablations of GriNNder consistently achieved significant speedups over HongTu. For smaller datasets, where host memory can store all intermediate activations and gradients, the configuration using only grad-engine activation regathering ('GRD-G') generally outperforms the storage-enabled version ('GRD-GC'), primarily due to cache management overhead. However, for larger datasets, employing storage alleviates host memory cache pressure, allowing the storage-based configuration ('GRD-GC') to deliver substantial speedups over both HongTu and GRD-G.

These results demonstrate that GriNNder is highly scalable for large datasets, with storage utilization being an effective strategy for handling large graphs on a single GPU. We also observed that GriNNder occasionally requires a larger number of partitions (i.e., different configurations) than HongTu. This is due to the GPU memory overhead introduced by overlapping GDS operations and computation. Despite this, GriNNder continues to deliver significant performance improvements over HongTu. It is also important to note that the number of partitions is merely a configuration hyperparameter, and users are not burdened by the need to manually handle this difference.

Table 10: **Training time/epoch (min) for various-sized Kronecker synthetic graphs.** '-' denotes when the number of partitions is not enough for running. **Bold** is the fastest training time in each (#layers, dataset) pair.

| #Layers | #Partitions | Method | 4.2M | 8.4M | 16.8M | 33.6M |
|---------|-------------|--------|------|------|-------|-------|
| 3 | 16 | HongTu | 0.43 | 0.83 | - | - |
| | | GRD-G | **0.29** | **0.59** | - | - |
| | | GRD-GC | 0.31 | 0.63 | - | - |
| | 32 | HongTu | 0.57 | 1.11 | 7.25 | - |
| | | GRD-G | 0.31 | 0.66 | - | - |
| | | GRD-GC | 0.33 | 0.71 | - | - |
| | 64 | HongTu | 0.76 | 1.76 | 10.70 | - |
| | | GRD-G | 0.41 | 0.77 | - | - |
| | | GRD-GC | 0.43 | 0.81 | - | - |
| | 128 | HongTu | 1.05 | 5.32 | 18.96 | 36.31 |
| | | GRD-G | 0.55 | 1.02 | **1.93** | **3.73** |
| | | GRD-GC | 0.58 | 1.05 | 1.99 | 3.86 |
| 5 | 16 | HongTu | 0.83 | 1.99 | - | - |
| | | GRD-G | **0.57** | **1.14** | - | - |
| | | GRD-GC | 0.60 | 1.20 | - | - |
| | 32 | HongTu | 1.07 | 8.04 | 19.15 | - |
| | | GRD-G | 0.60 | 1.30 | - | - |
| | | GRD-GC | 0.63 | 1.37 | - | - |
| | 64 | HongTu | 1.48 | 11.43 | 24.08 | - |
| | | GRD-G | 0.79 | 1.49 | - | - |
| | | GRD-GC | 0.84 | 1.55 | - | - |
| | 128 | HongTu | 4.61 | 17.08 | 37.09 | 96.99 |
| | | GRD-G | 1.08 | 1.96 | **3.71** | 10.87 |
| | | GRD-GC | 1.13 | 2.02 | 3.82 | **7.76** |

## O Cache hit rates

Table 11: **Cache hit rate**

| Dataset | Products | IGBM | Papers | kron-4.2M | kron-8.4M | kron-16.8M | kron-33.6M |
|---------|----------|------|--------|-----------|-----------|------------|------------|
| Hit Rate (%) | 28.57 | 53.70 | 83.63 | 80.81 | 80.47 | 92.77 | 92.70 |

We report cache hit rates in Table 11. As larger datasets (> IGBM, 10M) incur more reuse from the higher number of partitions, the hit rate is more significant in them. A low hit rate is natural in small datasets (e.g., Products) because we employ only a few partitions, and most data are not reused. Thus, GriNNder's caching is promising in large-graph training.

## P Comparison with existing lighweight partitioners

Table 12: **Time-to-quality comparison with spinner**

| | Products (4 parts) | | | | | | | | | | |
|---|------|------|------|------|------|------|------|------|---|---|---|
| Sec. | 0 | 1 | 2 | 3 | 4 | 5 | 6 | 7 | | | |
| Spinner | 2.62 | 1.98 | 1.78 | 1.47 | 1.23 | 1.20 | 1.19 | 1.19 | | | |
| GRD | 2.62 | 1.33 | 1.22 | 1.19 | 1.18 | | | | | | |
| | IGBM (32 parts) | | | | | | | | | | |
| Sec. | 0 | 1 | 2 | 3 | 4 | 5 | 6 | 7 | 8 | 9 | . . . | 37 |
| Spinner | 7.93 | 7.81 | 7.64 | 7.45 | 7.23 | 6.96 | 6.64 | 6.27 | 5.84 | 5.44 | . . . | 3.46 |
| GRD | 7.93 | 6.39 | 4.74 | 3.99 | 3.57 | 3.41 | 3.34 | 3.31 | | | |
| | Papers (2048 parts) | | | | | | | | | | |
| Min. | 0 | 2 | 4 | 6 | 8 | 10 | 12 | 14 | 16 | 18 | . . . | 23 |
| Spinner | 27.36 | 25.68 | 22.41 | 18.29 | 14.44 | 11.76 | 10.07 | 8.99 | 8.25 | 7.87 | . . . | 7.09 |
| GRD | 27.36 | 23.24 | 15.82 | 11.09 | 8.74 | 7.91 | 7.49 | 7.24 | 7.03 | 6.89 | | |

Table 13: **Comparison with SOTA out-of-core partitioner (2PS-L [64])**

| Quality/Time | Products | IGBM | Papers |
|--------------|----------|------|--------|
| 2PS-L | 2.08 / 210.19s | 5.20 / 202.77s | 18.39 / 86.56m |
| GRD | 1.18 / 4.00s | 3.31 / 6.96s | 6.89 / 17.60m |

We compared the time-to-quality (i.e., expansion ratio, $\alpha$, lower is better) of GriNNder's switching-aware partitioning (GRD) with the famous streaming algorithm (Spinner) in Table 12. We ran 50 iterations for both. We also benchmarked an out-of-core partitioner (2PS-L [64]) with the official code/settings in Table 13. GriNNder quickly results in higher-quality partitions for both cases.

## Q Convergence trend and practical overhead of switching-aware partitioning

Switching-aware partitioning converges fast with low practical overhead. In Table 14, we report the trend of the partitioning quality (score of the objective function) improvement (convergence) from the adjacent previous iteration (e.g., iter 4 → 5). We observe that at most 50 iterations are enough for convergence, thus limiting partitioning to 50 iterations in our experiments.

Given that a single iteration takes 0.08sec/0.14sec/21.12sec on average and our lightweight partitioning only requires 2.49sec/6.96sec/17.60min, partitioning consumes 0.07/0.02/0.39% of the total training time (500 epochs) on Products/IGBM/Papers, respectively.

Table 14: **Partitioning convergence trend**

| Dataset | Improvement (%) for Iterations | | | | | | | | | | |
|---|---|---|---|---|---|---|---|---|---|---|---|
| Products (4 parts) | Iteration | 1 | 5 | 10 | 15 | 20 | 25 | 28 (last) | | | |
| | Improve (%) | 6.81 | 9.75 | 3.79 | 0.36 | 0.12 | 0.08 | 0.05 | | | |
| IGBM (32 parts) | Iteration | 1 | 5 | 10 | 15 | 20 | 25 | 30 | 35 | 40 | 45 | 50 (last) |
| | Improve (%) | 11.13 | 7.78 | 3.66 | 1.96 | 0.66 | 0.77 | 0.39 | 0.21 | 0.16 | 0.10 | 0.08 |
| Papers (2048 parts) | Iteration | 1 | 5 | 10 | 15 | 20 | 25 | 30 | 35 | 40 | 45 | 50 (last) |
| | Improve (%) | 18.04 | 2.86 | 3.96 | 1.61 | 1.78 | 0.89 | 0.46 | 0.72 | 0.43 | 0.22 | 0.14 |

# R Configuration sensitivity results

Table 15: **Configuration sensitivity on training time (sec).** The default number of partitions for PRODUCTS and IGBM are 2 and 32, respectively.

| | | Method | ×1 | ×2 | ×4 | ×8 |
|---|---|---|---|---|---|---|
| 3-layer | PRODUCTS | HongTu | 9.98 | 11.11 | 12.22 | 13.65 |
| | | **GRD** | **6.93** | **7.72** | **8.55** | **8.99** |
| | IGBM | HongTu | 387.68 | 694.02 | 675.98 | 876.60 |
| | | **GRD** | **55.62** | **59.41** | **61.06** | **66.39** |
| 5-layer | PRODUCTS | HongTu | 19.14 | 21.46 | 23.42 | 26.22 |
| | | **GRD** | **13.65** | **15.22** | **16.38** | **17.60** |
| | IGBM | HongTu | 894.09 | 958.20 | 1183.88 | 1425.36 |
| | | **GRD** | **91.46** | **92.60** | **98.99** | **114.76** |

We additionally conducted configuration sensitivity experiments in Table 15. From the efficient caching management and elimination of redundancy, GriNNder is much less sensitive to the number of partitions (configurations). This enhances the practicality of GriNNder for end-users as they are not required to carefully configure the number of partitions.

# S Muti-GPU scalability

Although GriNNder was not designed for multi-GPU environments, it is scalable to some degree. We implemented multi-GPU GriNNder with partition parallelism and synchronization of scattered gradients in the backward pass. We ran this on a multi-GPU server with four RTX4090 GPUs*. Speedups of 1.25/1.60/2.44× and 1.23/1.53/2.14× were observed with 2/3/4GPUs, respectively, on IGBM and Papers. The speedup is proportional to the number of GPUs, where some overhead is incurred due to the system's shared resources – host memory bandwidth and storage bandwidth.

*: 2xIntel Xeon Gold 6442Y/512GB DDR5 DRAM/2TB PCIe5.0 NVMe SSD

# T Benchmarking w/o GDS

GriNNder can be generally used when GDS is unavailable. In this case, Kvikio (used in GriNNder) automatically switches to POSIX. Thus, users can still utilize GriNNder without any modification. Also, please note that GDS is supported on GPUs with NVIDIA compute capability >6.x (e.g., V100 and after).

We also benchmarked the performance (min) of GriNNder without GDS support in Table 16 as 'w/o GDS'. As Products and IGBM can be handled with host memory, the 'w/o GDS' performs similarly to the GDS cases. Even with Papers, where storage is highly utilized, there is only a 13-14% slowdown, demonstrating GriNNder's versatility.

Table 16: **Sensitivity to GDS**

| Layers | GDS | Products | IGBM | Papers |
|--------|-----|----------|------|--------|
| 3 layer | GDS | 0.12 | 0.93 | 9.07 |
| | w/o GDS | 0.12 | 0.93 | 10.25 |
| 5 layer | GDS | 0.23 | 1.52 | 12.03 |
| | w/o GDS | 0.23 | 1.52 | 13.73 |

## U  Cost efficiency analysis

Table 17 illustrates the cost efficiency of GriNNder compared to baselines. GriNNder is 33.26–60.71×
more cost-effective against distributed baselines and 6.97–9.78× cost-efficient than HongTu. Our
four server clusters cost \$131,848, including servers, 16 A6000 GPUs, and an Infiniband switch for
inter-server connection. Our single-GPU workstations cost \$3,300, including a workstation and an
RTX A5000 GPU. We calculate the vertex per second throughput and divide it by cluster/workstation
price to derive cost efficiency.

Table 17: **Cost efficiency (vertex per second / \$) of GriNNder compared to baselines.** We report
the cases runnable in Table 1.

| | | Method | PRODUCTS | IGBM | PAPERS |
|---|---|--------|----------|------|--------|
| $|L|=3$ | Dist. | CAGNET | 1.51 | 0.90 | 1.40 |
| | | SANCUS | 1.64 | 1.64 | - |
| | Limit. | HONGTU | 74.36 | 7.82 | - |
| | | **GRD** | **107.09** | **54.48** | **61.82** |
| $|L|=5$ | Dist. | CAGNET | 0.82 | 0.60 | - |
| | | SANCUS | 0.85 | 0.90 | - |
| | Limit. | HONGTU | 38.77 | 3.39 | - |
| | | **GRD** | **54.37** | **33.13** | **46.58** |

## V  Researches to resemble full-graph training with algorithm change

Many works have been proposed to resemble the accuracy (effect) of full-graph training by addressing
the information loss of mini-batch training. GNNAutoScale [25] utilizes staled activation to com-
pensate for the information loss of mini-batch training. LMC [79] further addresses the information
loss by compensating the information loss with gradients. In distributed full-graph training, many
researchers have tried to address the communication bottleneck while resembling the full-graph
training accuracy with staleness [90, 72] and error compensation [89] while proportional dropping of
communication. While the above compensation methods could be orthogonally applied to further
enhance the performance of GriNNder, we did not apply them to implement the exact full-graph
training without algorithm change.

## W  Functionality (accuracy) check of GriNNder

While GriNNder does not change the algorithm of full-graph training, we tested the accuracy of
GriNNder compared to full-graph training and HongTu for the functionality check, as illustrated in
Figure 17. Full-graph training was conducted with a CAGNET distributed baseline because Sancus
is not exact full-graph training. We only reported Products and IGBM because Papers was not
runnable on HongTu. As depicted in Figure 17, while HongTu is much slower than the distributed
setup, GriNNder provides significant speedup over the distributed CAGNET. All two baselines and
GriNNder show the same accuracy, which demonstrates the correct functionality of GriNNder. We

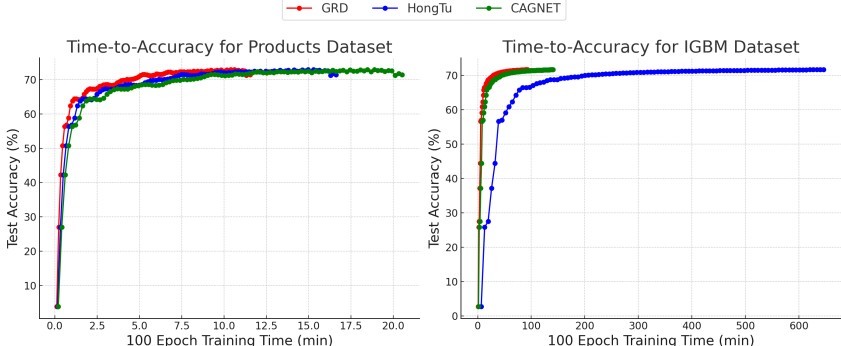

Figure 17: Functionality check of GriNNder.

also additionally checked Sancus's result, which is not the exact full-graph training as it utilizes staled activations and gradients. It shows similar accuracy to others but not exactly the same to them.

# X  Comparison with naïve baseline (naïve storage extension of ROC [42])

We also tested the naïve storage extension of ROC [42] instead of HongTu, which is the state-of-the-art framework. While we tested with HongTu with OS-based swap (i.e., `mmap`), we made it directly utilize storage instead of OS-based management for the ROC extension. On this naïve extension, GriNNder provides 1.28/29.00× speedup on 3-layer GCN on Products and IGBM, respectively. The speedup is significant on IGBM because Products only use #partitions=2 while IGBM uses #partitions=32. Thus, GriNNder provides further speedup on IGBM, which has much redundancy issue with ROC.

# Y  Limitations

## Y.1  Limitation of partition-wise cache management

Although we evaluate an extensive set of datasets and demonstrate the effectiveness of our partition-wise cache management, there can be a worst-case scenario: when dependencies are uniformly distributed across many partitions. In this case, partition-wise management may lead to overhead rather than performance improvement. We leave the handling of such a case to future work.

## Y.2  Discussion on SSD durability

A key concern when using SSDs for training is their lifespan, particularly due to durability issues. GriNNder is designed to minimize reliance on storage by leveraging host memory as much as possible. Specifically, when the host memory can accommodate all intermediate activations and gradients, GriNNder does not offload data to storage However, storage becomes necessary for large graph sizes or hidden dimensions. Since the write operations to SSDs are the primary factor impacting their lifespan, we can mitigate this issue by utilizing staled activations and gradients, as proposed in previous works [72, 90] from distributed full-graph training. By employing staleness techniques, while it is not the exact full-graph training, storage writes are effectively converted to storage reads, thereby extending the lifespan of the SSD. We plan to integrate these staleness-based techniques in an orthogonal manner to enhance the usability of `PyGriNNder`.

# Z  Related work

**GNN training.** Numerous methods have been proposed to learn representations from real-world graphs [38, 98, 80, 28]. Mini-batch training [33, 99, 103, 104, 46, 45, 96, 84, 60, 29, 57, 82] addresses memory constraints with sampling [107, 39, 20], but often exhibits input information loss [90, 42, 89, 45, 84]. Full-graph training is preferred for non input information loss (e.g., validating an algorithm's

sole effect), meeting memory requirements with many GPUs [90, 42, 89, 72, 85, 19, 91, 60, 93]. Near-memory processing is also adopted [106] as an alternative hardware solution. To enable full-graph training in a single server, [97] accumulates the weight gradients, and [92] stores activations/gradients to host memory. However, both are still limited to GPU or host memory capacity.

**SSD-based training.** Training DNNs with storage is a popular research area. Large language models, for example, [74] uses GPU, CPU, and SSD, and [40] additionally uses computational storage devices. But they are centered on managing optimizer states, which are extremely small compared to activations/gradients in full-graph training. Some works on mini-batch GNNs also utilize SSDs. Ginex [70] reduces I/O access by restructuring the GNN training pipeline and MariusGNN [88] loads only the valid graph features from storage with two-level partitioning. Helios [83] enables GPUs to directly access graphs in SSDs. DiskGNN [59] and GNNDrive [44] further optimize disk I/O of the above methods for mini-batching However, they target mini-batch training, and are limited by the message flow graph structure when extended to full-graph training.

**GNN snapshots.** Using snapshots is a popular method to reduce memory usage in DNN training [12, 7, 65, 32, 95, 66, 22] while providing exact results by storing activations and reconstructing them. Other strategies such as pruning [34, 24, 35, 36, 75, 58], quantization [16, 15, 17, 56, 105, 109, 77], and memory-efficient backpropagation [31, 30, 9] also reduce memory usage but may sacrifice accuracy. [101, 92, 8, 94, 43] also utilize snapshots for GNN training. [101] further reduces memory requirements and [92] naïvely stores snapshots of offloaded partitions, suffering from a huge redundancy. GriNNder instead proposes grad-engine activation regathering to address this redundancy and reduce the I/O overhead.

**Graph partitioning.** Partitioning is widely used for graphs [61, 92, 97, 47, 102, 41, 10, 100, 85, 60, 93, 91]. The popular METIS [47] features coarsening, partitioning, and un-coarsening phase [18, 37, 76, 5]. Additional frameworks [102, 41, 10] also try to balance partitions but demand a large amount of memory. Many distributed GNN training frameworks [90, 72, 103, 104, 60, 91, 93] are based on METIS for minimizing communication cost or workload balancing. For instance, [91] proposes an iterative METIS-based partitioning to enhance its three-dimensional parallelism. [50] reveals that previous partitioning [53] requires $4.8\times$-$13.8\times$ more memory than the graph itself. There are attempts to reduce this with online partitioning [21, 86, 81] or label propagation [48, 73] for scalable graph partitioning [63]. However, they focus on distributed systems and still require a lot of memory. Conversely, GriNNder proposes an efficient partitioning for large-scale graphs in limited environments.

