# OpenReview forum: "GriNNder: Large-Scale Full-Graph Training of Graph Neural Networks on a Single GPU with Storage"
_NeurIPS.cc/2025/Conference — Submitted to NeurIPS 2025_

### Official Review · Reviewer_W9jJ · 2025-06-24

**Clarity:** 2
**Significance:** 2
**Originality:** 3
**Rating:** 4
**Confidence:** 4

**Summary:**

This paper aims to efficently train full-graph GNN on single-GPU machines with the help of NVMe disks.  It conducts partition-level caching using the host RAM to reduce the I/O amplification, and designed a graph partitioning algorithm to reduce the cross-partition dependency.  It also designed a customized autograd engine tailored for its usage.

**Questions:**

1. Figure 3(a) and Figure 9 has the vertical axis being the number of required vertices from others.  But what matters for the partitioning algorithm is the number of required partitions, as the cache is performed in a partition-wise way.  I want to see an experiment to show the distribution of number of extra partitions required to show the effectiveness of the partitioning algorithm.
2. Relevant to the last question, won’t there be the long-tail effect so that the dependencies are spanned over many partitions?
3. I do not agree with the claim that performing partitioning on external servers would break the purpose of training GNNs on a single GPU.
   1. The partition can be done with a CPU-only server, and the partitioning only requires the graph structure, which will not be too large.  The training will still be performed in a single GPU.  Why is this not a feasible option?
   2. Figure 6 (a) seems questionable. The graph structure of papers100M is only tens of gigabytes (the whole dataset including features is 56GB). Why is METIS algorithm taking over 900GB on it?  I have previously run experiments using METIS on papers100M with a machine of 512GB.

**Ethical Concerns:**

["NO or VERY MINOR ethics concerns only"]

**Final Justification:**

Other than weakness 1, all concerns are addressed.  Weakness 1 requires a revision in the paper that more clearly states the modifications on the baseline systems in the main text and table.

**Limitations:**

Yes.

**Paper Formatting Concerns:**

1. There are both numbers for layer id and partition id in Figure 4.  It is confusing to follow the notations like A0 and $T_0$ where the 0 have different meanings.
2. Notations like GA0 are not explicitly defined; makes it harder to comprehend.

**Quality:**

3

**Strengths And Weaknesses:**

Strength:

This paper is a good paper: it made a lot of practical system designs and implementation.

Weaknesses:

1. The major flaw of this paper is the experiment design.  In the speed experiment (Table 1 and Table 6), time per epoch is compared.  However, it is problematic to directly compare the epoch time for full-graph training and mini-batch/sub-graph training, as the mini-batch methods will update the weight number of batch times per epoch, while the full-graph training will only update once, making full-graph training requires much more epoches to train and converge.  A reasonable metric between different training paradigms should be time to convergence (like Figure 12 in FreshGNN [1]).
2. The contents and style of this paper is more suitable for a system or data management conference.
3. A lot of large graphs are heterogeneous graphs, but heterogeneous GNNs like R-GCN are not considered in this paper.

[1] Huang, et al. FreshGNN: reducing memory access via stable historical embeddings for graph neural network training. VLDB 2024.

---

> ### Author Rebuttal · Authors · 2025-07-31
>
> We thank the reviewer for the valuable comments, detailed suggestions, and encouragement that our manuscript is a good paper with a lot of practical system designs and implementation. We have carefully addressed each concern below. **W$_{i}$**, **Q$_{j}$**, and **F$_{k}$** denote the weaknesses, questions, and formatting concerns, respectively.
>
> ### **[W1] Experiment design and metric clarification.**
>
> We sincerely apologize for the confusion caused by our insufficient explanation of experimental settings. To clarify, we believe the reviewer’s concern is not due to a design flaw, but rather due to unclear descriptions in our original manuscript. We used **time per epoch** as the metric because we compared full-graph training against **micro-batch training** (the **same training paradigm as full-graph training**), not mini-batch training.
>
> Specifically, we adapted all storage-aware mini-batch baselines (Ginex, DiskGNN, GNNDrive) to the micro-batch design (similar to Betty) for a fair comparison, as detailed in Section 5.1. This ensured each micro-batch:
>
> (i) covers all neighbors completely, and
>
> (ii) updates parameters **once per epoch**, identical to full-graph training.
>
> Thus, our experiments fairly compare runtime efficiencies under the same training paradigm as full-graph training.
>
> That said, we acknowledge that the reviewer may also be pointing out a valid concern: mini-batch training—which typically aims at fast convergence—can indeed be faster due to more frequent parameter updates. However, mini-batch training inherently sacrifices the completeness of graph information (i.e., introduces information loss), and this loss often becomes severe with additional optimizations to further reduce the mini-batch sampling overhead, as demonstrated in Fig. 2 of the reviewer-recommended reference (*FreshGNN*, VLDB 2024).
>
> Hence, mini-batch training and full-graph training represent distinct training paradigms, serving different goals. To avoid any confusion, we will explicitly clarify this distinction in the revised manuscript (Sections B and C). Additionally, since *FreshGNN* significantly reduces the aforementioned information loss with mini-batch training optimization, we will include this important and relevant paper in the final manuscript.
>
> ### **[W2] Conference fit.**
>
> Thank you for this valuable feedback. While we agree that system and data management venues could indeed be suitable, we intentionally chose NeurIPS as our primary venue because our contributions directly address significant needs of the ML research community, including:
>
> (i) A practical full-graph training framework enabling efficient GNN training on resource-constrained single-GPU systems.
>
> (ii) User-friendly, high-level APIs designed specifically to help ML practitioners quickly adopt efficient full-graph training without deep systems expertise.
>
> We believe that addressing ML-specific pain points aligns strongly with the NeurIPS audience, potentially shaping how large-scale GNN training and evaluation practices evolve within the community. To further emphasize this, we will extend the guide on API usage in the final manuscript to maximize accessibility to ML practitioners.
>
> ### **[W3] Lack of heterogeneous graph consideration.**
>
> Thank you for this suggestion. To address this, we have now extended GriNNder to explicitly support heterogeneous graph training (such as R-GCN). Our implementation involved creating a heterogeneous graph dataloader (*HeteroDataloader*). We validated this extension using the IGBM-hetero dataset with a two-layer heterogeneous convolutional model, consisting of a GCN layer (paper-cite-paper relation) and GraphSAGE layers (other relations). With hidden dimensions set to 128 (except for the output layer with 19 classes), GriNNder achieved 71.58% accuracy after 100 epochs, while significantly reducing runtime compared to HongTu (26.92 sec/epoch vs. 55.41 sec/epoch). We will include these additional experimental results and provide a clear implementation template in the revised manuscript, allowing easy extension to heterogeneous GNNs.
>
>
> ### **[Q1, Q2] Profiling partition-wise dependencies and the effectiveness of partitioning.**
>
> We sincerely thank the reviewer for the insightful comment. Partitioning remains essential, even when the partition dependencies span multiple partitions.
>
> *How many extra partitions are actually touched?*: On the Papers dataset (2048 partitions), a target partition depends on 1602.07± 234.11(Std.) other partitions, confirming the reviewer’s long-tail intuition.
>
> *Why does partitioning remain effective?*: This is because partitioning is significantly helpful in reducing host ↔ GPU traffic. Once partition-wise scheduling reduces random storage I/O and host-memory caching minimizes storage traffic, **host ↔ GPU traffic becomes the main bottleneck** (for details, please refer to **[Table C1]**). Without effective partitioning, the input activations required per partition—scaled by the **expansion ratio ($\alpha$)**—would dramatically increase host ↔ GPU traffic, severely harming performance.
>
> In detail, for every vertex that belongs to a partition, we need to fetch all dependent **extra** vertices through the host ↔ GPU path redundantly. The factor that affects the “extra fetching” is our expansion ratio ($\alpha$),
>
> – $\alpha = \frac{\sharp \text{vertices fetched}}{\sharp \text{vertices owned (for a partition)}}$
>
> Put differently, **$\alpha$ tells us how much larger the input tensor (to be sent between host ↔ GPU) becomes after we consider dependencies.** The total input-activation traffic for one partition is then
>
> – $\text{Traffic} = \alpha \times |H| \times N_{target}$,
>
> where $|H|$ is the hidden dimension and $N_{target}$ is the number of vertices the partition owns.
>
> **Partitioning is designed to drive $\alpha$ down** by co-locating vertices that are frequently accessed together as much as possible. In practice, this cuts the host ↔ GPU traffic relative to random partitioning (Table 5 in the manuscript), and delivers significant end–to-end speedup, even when a partition depends on many peer partitions. We will integrate this experiment (including a CDF of “# extra partitions per target partition”) and the above analysis into the final manuscript. These additions will make the role of the partitioner — and the significance of $\alpha$ — transparent to readers.
>
> **[Table C1] Training Time Breakdown of Storage-Offloaded Training**
>
> – **Storage-offloaded training with GriNNder optimizations**
> |Storage I/O|Host↔GPU I/O|Compute|Sync.|Etc.|
> |-|-|-|-|-|
> |19.2%| **48.4%** |8.6%|12.0%|11.8%|
>
> – **Storage-offloaded training without GriNNder optimizations**
> |Storage I/O|Host↔GPU I/O|Compute|Sync. & Etc.|
> |-|-|-|-|
> |85.4%|-|2.1%|12.5%|
>
> ##### For demonstration, we profiled a single epoch of storage-offloaded training of the IGBM dataset without/with GriNNder’s optimizations. Since GriNNder is highly optimized to overlap the compute and I/O, we employed the CPI stacking-like method$^+$ to break down the execution time approximately. When applying GriNNder optimizations, reduced random storage accesses from partition-wise I/O and minimized storage I/O from host memory caching make the host-to-GPU traffic the main bottleneck.
> ##### *$^+$: Eyerman et al., “A performance counter architecture for computing accurate CPI components,” ASPLOS, 2006.*
>
>
> ### **[Q3-1] Feasibility of external partitioning.**
>
> Thank you for raising this important point. We agree with the reviewer that offloading partitioning to an external CPU-only server is also feasible. Our intention was to emphasize practicality concerns: doing so adds data movement/management overhead, incurs extra cost, and breaks the simplicity of a “single‑machine” workflow, where some users may simply lack such infrastructure. Given the reviewer’s valid feedback, we will **soften lines 259-260 to “less practical”** rather than infeasible and clarify this nuance explicitly in the final manuscript.
>
> ### **[Q3-2] METIS memory usage in Fig. 6(a).**
>
> Thank you for the question. The >900GB peak memory usage stems from **MT-METIS** (the multi-threaded SOTA METIS), which is our default partitioner as discussed in Section 5.1. We utilized MT-METIS to achieve our goal of fast, end-to-end GNN training. To provide clarity, we report the peak memory usage of METIS and MT-METIS in **[Table C2]**. Due to the memory limit, we ran the experiment on a separate CPU server with a total of 48 cores and 96 threads (2x Intel Xeon Gold 6442Y) and 1TB of DDR5 memory. The results show that both METIS and MT-METIS exceed the host memory capacity of our default testbed (128GB). Additionally, we can verify that MT-METIS consistently consumes a significant amount of memory, similar to Fig. 6(a) in the manuscript. We will explicitly include these clarified results in the final manuscript.
>
> **[Table C2] Memory Usage (GB) and Partitioning Time (sec) of METIS and MT-METIS.**
> || 4 Parts  | 8 Parts | 16 Parts | 512 Parts | 1K Parts | 2K Parts |
> |-|-|-|-|-|-|-|
> | **METIS** | 367GB / 1914sec | 392GB / 2682sec | 368GB / 2638sec | 385GB / 4929sec | 385GB / 5233sec | 384GB / 5218sec |
> | **MT-METIS** | 1044GB / 875sec | 1009GB / 1005sec | 895GB / 907sec | 771GB / 1015sec | 755GB / 1067sec | 759GB / 1063sec |
>
>
> ### **[F1] Confusing Fig. 4 notation.**
> Thank you for pointing this out. We will update the notations in Fig. 4 accordingly. For example, we will modify $T_0$ to $T_{pt.0}$ to avoid confusion, and explicitly state that other numbers represent the layer ID.
>
> ### **[F2] Undefined notations.**
> We apologize for this oversight. We will explicitly define the notations in the final manuscript as follows:
>
> – $A^{l}_{pt.j}$ :  Output activation of a layer $l$ for a partition $j$.
>
> – $GA^{l-1}_{pt.j}$ : Gathered input activation of a layer $l$ for a partition $j$. It is gathered from the activation of layer $l-1$.
>
> – $T_{pt.j}$ : Topology of a partition $j$.

---

> > ### Comment · Reviewer_W9jJ · 2025-08-05
> > **Response to the author rebuttal**
> >
> > Thanks the authors for the rebuttal.  I am generally satisfied with the responses.  But I still have some concerns about the speed experiment.  Since the other systems like DiskGNN and GNNDrive are not designed for the full-graph training workflow, and the adapted version might be off by the original design in terms of performance, it should be clearly stated in the paper or table that these baselines are being adapted and how they are adapted.  I raise my score to weak accept.

---

> ### Author Response · Authors · 2025-08-05
>
> We sincerely thank the reviewer for reading our rebuttal and raising the score. We are pleased to hear that the reviewer was generally satisfied with our response. To address the concern, we will clearly state that baselines such as DiskGNN and GNNDrive are not designed for a full-graph training workflow, and the adaptation might be off from the original design regarding performance. Also, we will further clarify the adaptation of those baselines. Additionally, we will ensure to include all the contents of the response to the revised manuscript.

---

### Official Review · Reviewer_QQQV · 2025-07-03

**Clarity:** 3
**Significance:** 3
**Originality:** 4
**Rating:** 4
**Confidence:** 3

**Summary:**

This paper presents GriNNder, a full graph training pipeline leveraging activation/gradient offloading to NVMe SSD.

**Questions:**

- Is there an accuracy check on Paper (100M)?
- Would non-exact work that embrace the lossy compression side harder be relevant to this work? E.g., *EXACT: Scalable Graph Neural Networks Training via Extreme Activation Compression* ICLR 2022
- In the case that the model can be fully trained in one single GPU, is there any benefit of applying GRD to enable larger batch size or longer "sequence length"?

**Ethical Concerns:**

["NO or VERY MINOR ethics concerns only"]

**Final Justification:**

I am satisfied with the author's rebuttal and considered my raised concerns resolved. I am keeping my score as I believe it fairly reflects the contribution of the paper.

**Limitations:**

yes.

**Paper Formatting Concerns:**

no.

**Quality:**

3

**Strengths And Weaknesses:**

**Strengths**

* Large graph training faces different challenges than language models, where specific studies are welcomed. Lossless offloading also makes it much more likely to be adopted.
* Good coverage of scale to 100M+ graphs.
* Thorough report on system and efficiency metrics.

**Weaknesses**
* Lack of multi-GPU coverage with only a tiny amount of result in Appendix S. While I understand the work is not exactly proposed under this scenario, having multiple lower-grade GPUs is a typical scenario under a resource-constraint setting.
* Lack of fully in-single-GPU baseline to demonstrate the efficiency drop of GRD.
* (Minor cosmetic suggestion) Figures are too packed with minimum information provided in caption, readers would need to jump between the anchoring sections and the figures to grasp the message.

---

> ### Author Rebuttal · Authors · 2025-07-31
>
> We thank the reviewer for the insightful comments and constructive suggestions. We especially appreciate the reviewer for recognizing our good scalability, lossless design, and thorough evaluations. We carefully addressed each point raised, where **W$_{i}$** and **Q$_{j}$** denote the weaknesses and questions, respectively.
>
> ### **[W1] Lack of multi-GPU coverage.**
> Thank you for highlighting the importance of multi-GPU scenarios. While GriNNder primarily targets single-GPU training, we acknowledge that multi-GPU setups are also widely used in resource-constrained environments. To address this, GriNNder already supports multi-GPU training (Section S), and we will further provide a more detailed discussion in the final manuscript as follows. Specifically, we will:
>
> – Provide detailed descriptions of our **multi-GPU implementation**.
>
> – Add experimental comparisons with the state-of-the-art baseline, HongTu (**[Table B1]**).
>
> – Provide a user-friendly open-source API supporting multi-GPU configurations upon release.
>
>
> Our multi-GPU extension uses two lightweight mechanisms:
>
> **(i) Partition parallelism**: We divide the partitions into disjoint #(GPU) sets; each GPU performs forward/backward on its set independently.
> **(ii) Weight/Gradient synchronization**: During the backward pass, partial gradients of dependent vertices from different GPUs are atomically accumulated on the host, which accumulates the gradients of the vertices. Before the weight update, a weight gradient all-reduce operation is conducted between GPUs to synchronize the weights among them.
>
> This simple yet effective approach achieves excellent scalability with minimal overhead, significantly outperforming HongTu, as illustrated in **[Table B1]**.
>
> **[Table B1] Multi-GPU Scalability Comparison (Normalized to GriNNder-1GPU performance)**
>
> |IGBM|1GPU|2GPU|3GPU|4GPU|
> |-|-|-|-|-|
> |HongTu|0.17x|0.21x|0.27x|0.34x|
> |GriNNder|1.00x|1.25x|1.60x|2.44x|
>
> |Papers|1GPU|2GPU|3GPU|4GPU|
> |-|-|-|-|-|
> |HongTu|Swap OOM|Swap OOM|Swap OOM|Swap OOM|
> |GriNNder|1.00x|1.23x|1.53x|2.14x|
>
> *Note: For the multi-GPU implementation of HongTu, we applied a similar multi-GPU extension with GriNNder, the details of which are mentioned in the response.*
>
> ### **[W2] Lack of in-single-GPU baseline.**
>
> Thank you for the insightful feedback. GriNNder does not incur an efficiency drop when the entire graph fits in GPU memory, since GriNNder detects this case and automatically bypasses host/SSD offloading, resorting to pure GPU or GPU-host execution. However, to explicitly evaluate the potential overhead of our approach, we conducted additional experiments by forcibly enabling host/storage offloading on smaller datasets that fit within a single GPU's memory.  We compare the vanilla PyG run (which GriNNder will automatically choose, PyG/GRD), GriNNder with *forced* host memory offloading (GRD-Forced Host), GriNNder with *forced* storage offloading (GRD-Forced Storage), and HongTu in **[Table B2]**.  While forced offloading inevitably introduces overhead, GriNNder’s optimizations substantially minimize efficiency loss compared to HongTu, clearly demonstrating the effectiveness of our method.
>
> **[Table B2] Training Time per Epoch (seconds) Comparison for In-Single-GPU Datasets.**
> |            | PyG/GRD  | GRD-Forced Host  | GRD-Forced Storage | HongTu |
> |------------|------|------|-------------|--------|
> | **ogbn-arxiv** | 0.21 | 0.49 | 0.65        | 0.67   |
> | **Reddit**     | 0.71 | 1.10 | 1.28        | 1.55   |
>
> We will include these results and detailed discussions explicitly in the revised manuscript.
>
> ### **[W3] Packed figures and captions.**
> Thank you for this suggestion. We apologize for any inconvenience. In the revised manuscript, we will improve the readability and visibility of figures by increasing spacing, enhancing clarity, and enriching the captions to ensure each figure’s message is self-contained and clear.
>
> ### **[Q1] Accuracy on Papers dataset.**
> Thank you for this inquiry. On the Papers dataset, GriNNder achieves identical accuracy (63.04%) as CAGNET. We had initially excluded this accuracy result from Section W in the manuscript because HongTu consistently encountered out-of-memory issues. We will explicitly include this result in the revised manuscript.
>
> ### **[Q2] Relevance of lossy compression.**
> Thank you for pointing out an important reference (*EXACT*, ICLR 2022). While GriNNder focuses explicitly on lossless offloading, we acknowledge that lossy compression techniques could further reduce I/O overhead (i.e., applying activation/gradient quantization for storage-offloaded data), complementing our framework. We will explicitly discuss this complementary perspective and include this reference in Section V of our revised manuscript.
>
> ### **[Q3] Applicability of GriNNder on larger batch sizes/longer sequences.**
> Thank you for the insightful question. Yes, GriNNder enables much larger batch sizes or longer sequence lengths. In GNN training, supporting a larger batch size or longer sequence length means that the increase in the number of nodes can be handled. As illustrated in Table 1 in the manuscript, GriNNder enables scaling up to datasets with a larger number of nodes, while OOM limits other baselines. This advantage is also demonstrated when the hidden sizes are increased (Table 3 in the manuscript). If our interpretation of "sequence length" as graph size differs from the reviewer’s intention, please let us know, and we would be happy to clarify further.

---

> > ### Comment · Reviewer_QQQV · 2025-08-02
> >
> > I am satisfied with the rebuttal and will maintain my score.
> >
> > I also checked reviewer `W9jJ`'s comment. While I will left the technicality parts to the authors and that reviewer, I want to weigh-in and note that a framework-like contribution is, in my opinion, perfectly suitable to a conference like NeurIPS. While many ML framework do opt to publish in the more system end of conferences (OSDI, SOSP, etc.), prior art like FlexGen and DeepSpeed-MoE are published in ICML/ICLR-like conference, so the tradition is strong.

---

> > > ### Author Response · Authors · 2025-08-03
> > >
> > > Thank you for reading our rebuttal. We are pleased to hear that the reviewer was satisfied with our response. Also, thank the reviewer for emphasizing the venue fit of our manuscript for NeurIPS. In the revised manuscript, we will ensure that all content is included in the response.

---

### Official Review · Reviewer_ttmn · 2025-07-05

**Clarity:** 3
**Significance:** 3
**Originality:** 3
**Rating:** 4
**Confidence:** 4

**Summary:**

Full-graph GNN training suffers severe scalability issues. GriNNder considers a resource-constraint scenario where training data (e.g. activation values) has to be stored on disk. A new partition algorithm and caching method are proposed. Although many of these approaches are well-studied (e.g., caching, graph partition, redundancy elimination), the integration and full-stack system are the core contributions.

**Questions:**

- Why accelerating single-GPU training is an important research topic?
- How is multi-GPU training implemented? Section S provides brief introduction but it is hard for me to understand the details.

**Ethical Concerns:**

["NO or VERY MINOR ethics concerns only"]

**Final Justification:**

I appreciate authors' additional comment, and hope the authors to include them in the revision. I maintain my positive score.

**Limitations:**

Yes

**Quality:**

3

**Strengths And Weaknesses:**

## Strengths

- A multi-level caching to optimize I/O. A new caching policy is developed.
- A new elimination method to avoid storage redundancy.
- A new lightweight partition algorithm is developed.
- User-friendly API design is provided.
- Comprehensive evaluation.

## Weaknesses

- The authors claim that GriNNder achieves comparable performance with distributed GNN training. However, Sancus and CAGNET are inappropriate baselines for evaluating distributed GNN training because they use broadcast for data transfer which incurs severe redundant communication. Increasing GPU numbers decreases their efficiency. These experiments are misleading as they do not use the best setting of distributed GNN training. Nonetheless, this is not a killing point as this paper focuses on single-GPU training.

- Nonetheless, it is questionable whether accelerating single-GPU training is an important research question. A multi-GPU cluster for ML training are affordable to most academia group and industry companies nowadays.

- Redundancy-free GNN was studied in [1], but this related work is not discussed.

[1] Redundancy-Free Computation Graphs for Graph Neural Networks

---

> ### Author Rebuttal · Authors · 2025-07-31
>
> We thank the reviewer for the thoughtful comments and insightful questions. We are also grateful for the reviewer recognizing the novelty in our caching design, redundancy elimination methods, lightweight partitioning algorithm, user-friendly API, and comprehensive evaluations. We address each of the reviewer’s points below, where **W$_{i}$** and **Q$_{j}$** denote the weaknesses and questions, respectively.
>
> ### **[W1] Concerns related to the distributed baselines.**
> Thank you for raising the concern about the selected baselines. As stated in the manuscript (lines 304-305), while the Products dataset can be run on fewer GPUs (other datasets require all 16 GPUs), we maintained the 16-GPU hardware setting for a consistent evaluation environment. We have specifically included Sancus, although it is not an ideal baseline, because it addresses the communication bottleneck inherent in multi-GPU setups by reusing stale gradients to reduce inter-GPU communication. To address the reviewer’s concern clearly, we will explicitly highlight the limitations of these distributed baseline comparisons in the revised manuscript.
>
> ### **[W2, Q1] Importance of single-GPU training.**
> We appreciate the reviewer sharing this critical point. While we agree that multi-GPU setups are common, accelerating single-GPU training remains a valuable and actively researched problem, particularly due to three practical considerations:
>
> **(i) Access & reproducibility**:
> Not all academic or industrial groups have consistent access to GPU clusters, as discussed extensively in our survey (Appendix A). Single-GPU training reduces resource barriers, democratizes research, and simplifies reproducibility by allowing results to be verified without large infrastructures.
>
> **(ii) Limitations of distributed full-graph training**:
> Distributed full-graph training often hits severe **communication bottlenecks** due to synchronization overhead for features, activations, and gradients across multiple GPUs. A single storage-augmented GPU approach avoids these inter-GPU communication costs altogether, substantially reducing complexity and runtime.
>
> **(iii) Lessons from adjacent ML domains**:
> Recent trends in large language model (LLM) research emphasize storage-augmented single-GPU training. For instance, FlexGen enables single-GPU inference of massive models (100B+ parameters) via host memory offloading. Similarly, QLoRA and ZeRO-Infinity leverage single-GPU setups augmented with storage to scale training beyond conventional GPU limits. We believe these examples demonstrate the practicality and research importance of efficient single-GPU strategies.
>
> Moreover, GriNNder **supports multi-GPU environments** (Section S). To clarify this contribution, we will explicitly emphasize this extension in the revised manuscript, along with detailed descriptions and open-source APIs to facilitate usage.
>
> ### **[W3] Missing related work.**
> Thank you for pointing out this oversight. The suggested paper (*Redundancy-Free Computation Graphs for Graph Neural Networks*, KDD 2020) addresses redundant computation within message-passing structures by introducing hierarchically aggregated computation graphs (HAGs). In contrast, GriNNder specifically targets I/O optimization by eliminating redundant storage of activations and gradients needed for full-graph training, especially considering layer-wise dependencies and storage constraints. Thus, the goals and technical approaches of the two works are complementary but distinct. We will explicitly include and clearly discuss this related work in the revised manuscript.
>
> ### **[Q2] Clarification of multi-GPU training implementation.**
> Thank you for requesting clarification. Our multi-GPU implementation employs two lightweight mechanisms:
>
> **(i) Partition parallelism**: We divide the partitions into disjoint #(GPU) sets; each GPU performs forward/backward on its set independently.
> **(ii) Weight/Gradient synchronization**: During the backward pass, partial gradients of dependent vertices from different GPUs are atomically accumulated on the host, which accumulates the gradients of the vertices. Before the weight update, a weight gradient all-reduce operation is conducted between GPUs to synchronize the weights among them.
>
> These straightforward enhancements enable effective multi-GPU execution with minimal overhead. We will incorporate these details clearly into the revised manuscript.

---

> > ### Author Response · Authors · 2025-08-06
> >
> > Thank you again for your thoughtful review and constructive feedback on our manuscript. We have posted a detailed rebuttal addressing your comments. We would be very grateful if you could kindly review our responses and let us know whether the concerns have been addressed. Your insights would be valuable in helping us further improve the clarity and impact of our work. Thank you again for your time and efforts.

---

> > > ### Comment · Area_Chair_JuJh · 2025-08-08
> > > **[URGENT] please response to rebuttal**
> > >
> > > Dear Reviewer ttmn,
> > >
> > > You have not participated in the discussion of the rebuttal. Please be reminded that there would be consequences for not being responsive during the rebuttal phase. We would request you to please study the rebuttal posted by the authors and share your assessment.
> > >
> > > best,
> > >
> > > AC

---

### Note · Authors · 2025-08-12

We thank the reviewers for their thoughtful and constructive discussions and the AC for encouraging engagement. GriNNder enables high-throughput and large-scale full-graph training with a single GPU, utilizing storage devices, achieving **up to 9.78x** speedup.

***

### **[Rebuttal summary]**

Following the rebuttal, **all participating reviewers expressed satisfaction with our responses**. Reviewers increased/maintained positive ratings, with positive recommendations and no new technical objections. Reviewer (hereafter R) `QQQV` further emphasized that our framework-style system contributions have a strong **venue fit for NeurIPS**, similar to FlexGen/DeepSpeed-MoE.

***

### **[Key strengths from the reviewers]**

1. Novel multi-level caching, redundancy-free storage, lightweight partitioning.

2. Comprehensive evaluations.

3. Practical and user-friendly API for ML practitioners.

4. Clear venue fit for NeurIPS, as affirmed by R `QQQV`.

***

### **[Rebuttal items]**

1. **Distributed baselines**: We clarified limitations of distributed baselines (R `ttmn`).

2. **Single-GPU importance**: We reinforced its relevance by highlighting broader accessibility, easier reproducibility, avoidance of inter-GPU bottlenecks, and parallels to recent single-GPU LLM work (e.g., FlexGen, QLoRA, ZeRO-Infinity) (R `ttmn`).

3. **Multi-GPU details**: We added detailed design/experiments showing superiority over the baseline (R `ttmn/QQQV`).

4. **Baseline adaptation**: We stated that DiskGNN/GNNDrive are not initially designed for full-graph training, described our micro-batch adaptations, and noted potential performance impact (R `W9jJ`).

5. **Heterogeneous GNNs**: We extended GriNNder to support heterogeneous GNNs with significant speedups (R `W9jJ`).

6. **Partitioning analysis**: We added a new insight with dependency CDFs, long-tail confirmation, and explanation of host↔GPU traffic reduction (R `W9jJ`).

7. **Other improvements**: We clarified captions/notations, softened language on external partitioning, added the Papers dataset accuracy, discussed suggested relevant works, and noted no efficiency drop for in-single-GPU cases.

***

We believe these clarifications/additions address all concerns and strengthen the manuscript. We will incorporate them in the revised version and hope the AC will view GriNNder as a timely, solid contribution to large-scale full-graph GNN training for limited resources with a clear impact and strong fit for the NeurIPS community.

---

### Decision · Program_Chairs · 2025-09-17

**Decision:**

Reject

**Comment:**

The paper introduces an algorithm for training GNNs on large graphs using a single GPU with sufficient storage. It incorporates several system-level innovations that achieve significant speed-ups (up to ~9X) on empirical benchmarks. These innovations include a multi-level caching policy for optimized I/O, an elimination method to reduce storage redundancy, and a lightweight partitioning algorithm.

All reviewers rated the paper positively with a score of 4, but no one is championing it due to concerns about its suitability for ICLR, as the focus is more on system-level aspects than on a machine learning algorithm. Overall, the paper might be a better fit for SIGMOD, VLDB, or KDD.